# Discovering allatostatin type-C receptor specific agonists

Kübra Kahveci[1], Mustafa Barbaros Düzgün[1], Abdullah Emre Atis [2], Abdullah Yılmaz[2], Aida Shahraki[1,3], Basak Coskun [2], Serdar Durdagi[4,5,6] ✉ & Necla Birgul Iyison[1] ✉

Currently, there is no pesticide available for the selective control of the pine processionary moth (*Thaumetopoea pityocampa*-specific), and conventional methods typically rely on mechanical techniques such as pheromone traps or broad-spectrum larvicidal chemicals. As climate change increases the range and dispersion capacity of crop and forest pests, outbreaks of the pine processionary occur with greater frequency and significantly impact forestry and public health. Our study is carried out to provide a *T. pityocampa*-specific pesticide targeting the Allatostatin Type-C Receptor (AlstR-C). We use a combination of computational biology methods, a cell-based screening assay, and in vivo toxicity and side effect assays to identify, for the first time, a series of AlstR-C ligands suitable for use as *T. pityocampa*-specific insecticides. We further demonstrate that the novel AlstR-C targeted agonists are specific to lepidopteran larvae, with no harmful effects on coleopteran larvae or adults. Overall, our study represents an important initial advance toward an insect GPCR-targeted next-generation pesticide design. Our approach may apply to other invertebrate GPCRs involved in vital metabolic pathways.

Broad-spectrum pesticides are regularly employed at an industrial scale to manage pests, prevent disease, and safeguard against crop failure[1]. They have become an inevitable part of modern life such that every year around 3 billion kg of pesticides are used worldwide[2]. Extensive and improper usage of these chemicals gave rise to severe problems for the environment and human health[3]. Consequences of soil, water, and food contamination eventually lead to pest resistance, environmental toxicity, and bioaccumulation, threatening biodiversity and public health[4,5]. Several studies reported that chronic exposure to some pesticides could increase the reactive oxygen species levels which is correlated with genotoxicity, inflammatory response, cancer, and neurological disorders like Parkinson's disease[6–8].

Despite their significant drawbacks, the benefits provided by pesticides should not be denied. Pest-management methods are a fundamental component of the third agricultural revolution, which resulted in affordable food, crop yield reclamation, and economic benefits such as chemical feedstocks and renewable fuel supplies[9]. Furthermore, pesticides contribute to the prevention of vector-borne diseases and have been used to control epidemic outbreaks[10]. Consequently, there is an increasing demand for safer and target-specific compounds that minimize unwanted side effects while maintaining the benefits of pesticide use.

Novel approaches have been developed to circumvent issues such as pest resistance and environmental toxicity, including the use of biological agents (e.g., bacteria)[11]. Secondary metabolites, RNA interference, juvenile hormone (JH)-mimetic peptides, neuropeptides, and their corresponding receptors have also been implemented as next-generation pesticides[12–14]. Neurohormone and neuropeptide G protein-

[1]Department of Molecular Biology and Genetics, Boğaziçi University, İstanbul, Türkiye. [2]Plant Protection Product and Toxicology Department, Plant Protection Central Research Institute, Ankara, Türkiye. [3]Kolb Lab, Department of Pharmacy, The Philipp University of Marburg, Marburg, Germany. [4]Molecular Therapy Lab, Department of Pharmaceutical Chemistry, School of Pharmacy, Bahçeşehir University, İstanbul, Türkiye. [5]Computational Biology and Molecular Simulations Laboratory, Department of Biophysics, School of Medicine, Bahçeşehir University, İstanbul, Türkiye. [6]Lab for Innovative Drugs (Lab4IND), Computational Drug Design Center (HITMER), Bahçeşehir University, İstanbul, Türkiye. ✉e-mail: serdar.durdagi@bau.edu.tr; birgul@bogazici.edu.tr

coupled receptors (GPCRs) are particularly appealing targets in this context, due to their roles in regulating a wide array of physiological processes in both vertebrates and invertebrates[15]. These properties also make them a topic of great interest for drug discovery efforts: around one-third of clinically approved drugs target GPCRs, with class-A GPCRs (94% of available GPCR drugs) and small molecule drugs (92%) accounting for the majority of research[16].

Allatostatin receptors (AlstRs) are neuropeptide-activated class-A GPCRs widely expressed in the neuroendocrine systems of insects[17]. They exhibit similarities with the human somatostatin and opioid receptors[18]. Their primary role is to inhibit the production of JHs, multi-functional regulatory terpenes secreted from endocrine glands called *corpora allata* (CA) that are typically adjacent and posterior to the insect brain[19]. Due to their central role in mediating insect reproduction, molting, pupation, ecdysis, behavior, diapause, and stress responses, the synthesis and regulation of JHs have been extensively characterized through molecular, biochemical, and -omics methods. Allatostatin peptides (ASTs) are particularly important for JH regulation and are found in the central nervous system and neurohemal areas of insects and crustaceans[20]. Although three families of allatostatins are currently recognized, some have lost their role as inhibitors of JH synthesis across insect orders[21]. For example, class-A allatostatins (also called FGLa allatostatins) have only been shown to downregulate JH production in cockroaches, termites and crickets; while regulating feeding decisions, growth, and metamorphosis in *Drosophila*[22]. Similarly, B-type allatostatins play multiple physiological roles in various arthropods, influencing ecdysis behavior, the circadian clock, feeding, locomotion, and reproductive physiology. They are especially prominent as inhibitors of muscle contractions, and are also known as myoinhibiting peptides (MIPs) in this role[23]. The largest family Allatostatin A (i.e, FGLa allatostatins) are isolated from cockroach *Diploptera punctata*. The sequence Tyr/Phe-Xaa-Phe-Gly-Leu-amide in the C-terminal is conserved among the family[24]. Despite the conservation of sequence, the role of the AST-A peptides as inhibitors of JH production has only been maintained in selected insect species such as cockroaches, crickets, and termites. The studies in *Drosophila melanogaster* showed that AST-A peptides regulate feeding decisions, growth, and metamorphosis[21].

First discovered in the cricket *Gryllus bimaculatus*, B-type allatostatins are characterized by a conserved C-terminus W(X)$_6$Wamide. Genome projects have uncovered AST-B sequences in numerous insects, crustaceans, and the centipede *Strigamia maritima*. In various arthropods, B-type allatostatins play multiple physiological roles, influencing ecdysis behavior, the circadian clock, feeding, locomotion, and reproductive physiology. The peptides share a similar sequence with a MIP. In most insects, the AST-Bs act as MIPs, serving as inhibitors of the contraction of the visceral muscle[25].

Allatostatin-C peptides (AST-Cs) were first identified and shown to inhibit JH production in *Manduca sexta*, while homologs were later found in mosquitoes and moths. The Pro-Ile-Ser-Cys-Phe (PISCF) sequence is characteristic of the family which appears to be the only allatostatin type to inhibit JH synthesis in Lepidoptera[26]. AST-C has N-terminal pyroglutamic acid blocking and a disulfide bridge is located between Cys7 and Cys14[27]. *Bombyx mori* AST-C inhibits JH biosynthesis at every developmental stage, contrasting with *M. sexta* where it is active only in fifth instar larvae and adult females[28]. A study in the mosquito *Aedes aegypti* suggests blockage of citrate transporters in mitochondria to limit acetyl-CoA in the cytoplasm for JH synthesis as a possible action mechanism for JH inhibition by AST-C[29]. Beyond JH regulation, AST-Cs in *D. melanogaster* function as immunosuppressive peptides after bacterial infection, preventing premature cell death. Additionally, they serve functions like myoinhibition, heart muscle contraction, and modulation of circadian activity depending on the organism[30].

Although the exact mechanism remains unclear, it is well-established that the allatostatin peptide family can inhibit JH synthesis in a robust, rapid, and reversible manner[31,32]. As such, due to their significant role in the regulation of insect development, AlstRs are potential targets for pest control agents.

*Thaumetopoea pityocampa* (*T. pityocampa*), also known as the pine processionary moth, is a pest species that feeds on pine needles and limits pine forest development[33]. The caterpillar of this species also possesses urticating hairs that may cause serious health issues in humans and pets, typically presenting as skin rash, urticaria, or ocular and respiratory conditions arising from contact with airborne hairs[34–36]. Although the native range of the pine processionary is limited to temperate Mediterranean forests, global warming has allowed the establishment of populations further north[37]. Existing ways for countering the *T. pityocampa* include physically or chemically destroying egg batches and communal nests (also known as silk nests or bivouacs built by the caterpillars) made from silk threads and pine threads at the crown of the trees, exposed to the sun to keep the colony warm[38]. Pheromone traps, essential oils, microbial agents, and the natural predator *Calosoma sycophanta* insects are also employed[39–42]. However, all these approaches are only partially successful, necessitating the development of a more potent and specific pesticide since there is no *T. pityocampa*-specific pesticide available yet. While pesticides such as diflubenzuron have been used against *T. pityocampa*, these agents have large target spectra and off-target side effects against endangered species[43]. Consequently, the development of a *T. pityocampa*-specific pesticide would significantly advance control efforts against this emerging pest.

In this work, to provide a *T. pityocampa*-specific pesticide, we target the orthosteric site of the AST-C receptor (AlstR-C) of this species. We use a virtual screening approach to screen two small molecule databases and identify potential hit compounds. We further analyze selected hit compounds that exhibit strong interactions with AlstR-C active site through molecular dynamics (MD) simulations and molecular mechanics generalized Born surface area (MM/GBSA) binding free energy calculations. We perform biological evaluation of candidate molecules and the AST-C peptide through a TGF-α shedding assay, while we evaluate lethality and side effects in vivo on larvae and adult insects. Figure 1 represents the workflow of applied procedures. Our combinational studies reveal four agonists exhibiting suitable properties as novel potential pesticide to counteract the pine processionary moth and contribute to future next-generation pesticide research.

## Results

### Virtual screening of small molecule databases leads to the identification of novel hit molecules

ChemDiv GPCR-Targeted (64515 compounds) and Peptidomimetic (61518 compounds) libraries were used in virtual screening, targeting the orthosteric pocket of the AlstR-C. Since no crystal structure exists for the AlstR-C protein, studies were conducted using a homology-modeled structure previously developed by our research group[44]. Grid generation was performed on the docking pose of the AST-C peptide. After docking simulations, a total of ten molecules were chosen from the two libraries for further analysis based on docking scores (Fig. 2). Among these hit compounds, D074-0034 molecule was identified to have the highest docking score (Supplementary Fig. 1, Supplementary Table 1).

### Analyses of MD simulations help to understand better the interactions of identified hit molecules in the orthosteric site

MD simulations were applied on the protein-ligand complexes of the top-10 hit molecules (Supplementary Table 2) with AlstR-C for 100 ns to estimate the structural stabilities of the complexes and interactions within the binding cavity. Three replicates of simulations (100 ns × 3) were conducted for each system. Trajectory frames collected during

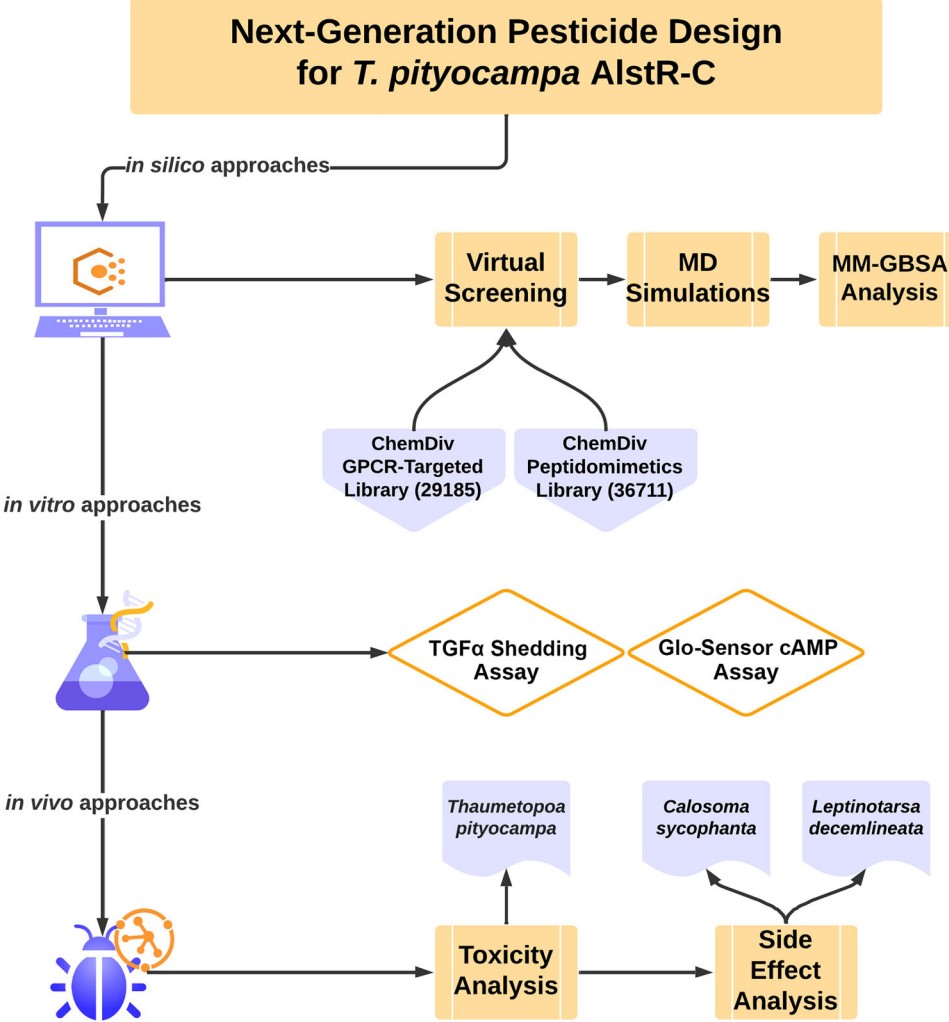

**Fig. 1 | Workflow of the study.** Schema indicates the research approaches for the next-generation pesticide design. It combines in silico approaches (i.e., virtual screening, MD simulations, and post-MD analyses), in vitro biological assays (i.e., TGFα shedding assay, and Glo-Sensor cAMP assay), and in vivo approaches (i.e., toxicity and side effect analysis).

the simulations were analyzed with the Simulation Interaction Diagram module of Desmond. This analysis included root-mean-square-deviations (RMSD) and root-mean-square-fluctuations (RMSF) of Cα atoms and protein-ligand contacts. Assessment of Cα RMSD is commonly used to observe complex structural stability[45]. Trajectory analysis revealed that protein-ligand complexes reached equilibrium after 10 ns. However, all complexes did not show the same structural stability. Cα plots demonstrated RMSD values ranging from 2 to 4 Å (Supplementary Fig. 2a). Significant variance was observed for Lig-fit-Protein RMSDs, indicating distinct stabilities in each ligand-protein complex (Supplementary Fig. 2b). Some compounds achieved equilibrium after a longer simulation duration and exhibited conformational changes compared to the ligand's reference position.

RMSF calculations can provide insights into the dynamic behavior of the residues[46]. RMSF plots of all the ligands with AlstR-C are shown in Supplementary Fig. 3. The residues with the highest fluctuations are regions of the extracellular and intracellular loops, N-terminal and C-terminal domains. For all plots, most of the residues have fluctuations within 2.5 Å. The fluctuation patterns of the residues in the complexes are comparable.

Simulation trajectories were examined to evaluate the dynamics between the receptor and ligands. The sulfonyl group of C300-0328 interacted with Tyr113 by 28% through water bridges and Cys187 of the receptor through hydrogen bonds and water bridges over 75% of the

simulation time (Supplementary Fig. 4a). Gln278 was next mostly interacted residue along with Asn183, Ser186, and Asn188, which were involved in hydrogen bonds and water bridges throughout 50% of the simulation time. Gln278 established hydrogen bonds and hydrophobic interactions with the nitrogen atoms of phenylpiperazine and propenamide groups. The π-π stacking interaction was noted with the Trp191 residue, whereas Phe291 contributed with hydrophobic interactions (Supplementary Fig. 4a).

Compared to the C300-0328, compound C794-1617 built a smaller number of noncovalent interactions with the receptor, and these contacts were not continuous for most of the simulation time (Supplementary Fig. 4b). Two of the most prominent were the hydrophobic interactions with Leu274 and polar contacts of the amide group with Thr280.

The sustained interactions were significant in the D074-0013:AlstR-C complex, and polar contacts through hydrogen bonds and water bridges were significant. Interactions with Asn188, located at the ECL-2, were remarkable since this residue was in contact with the ligand for 87% of the simulation time (Supplementary Fig. 4c). Gln271 was also notable because it sustained its interactions for 80% of the simulation time. It must be noted that this residue was identified as vital for the G-protein-dependent activation of the AlstR-C[44].

The ligand (D074-0034) was the analog of the D074-0013, and this structural similarity was observable in the contacts. Hydrogen

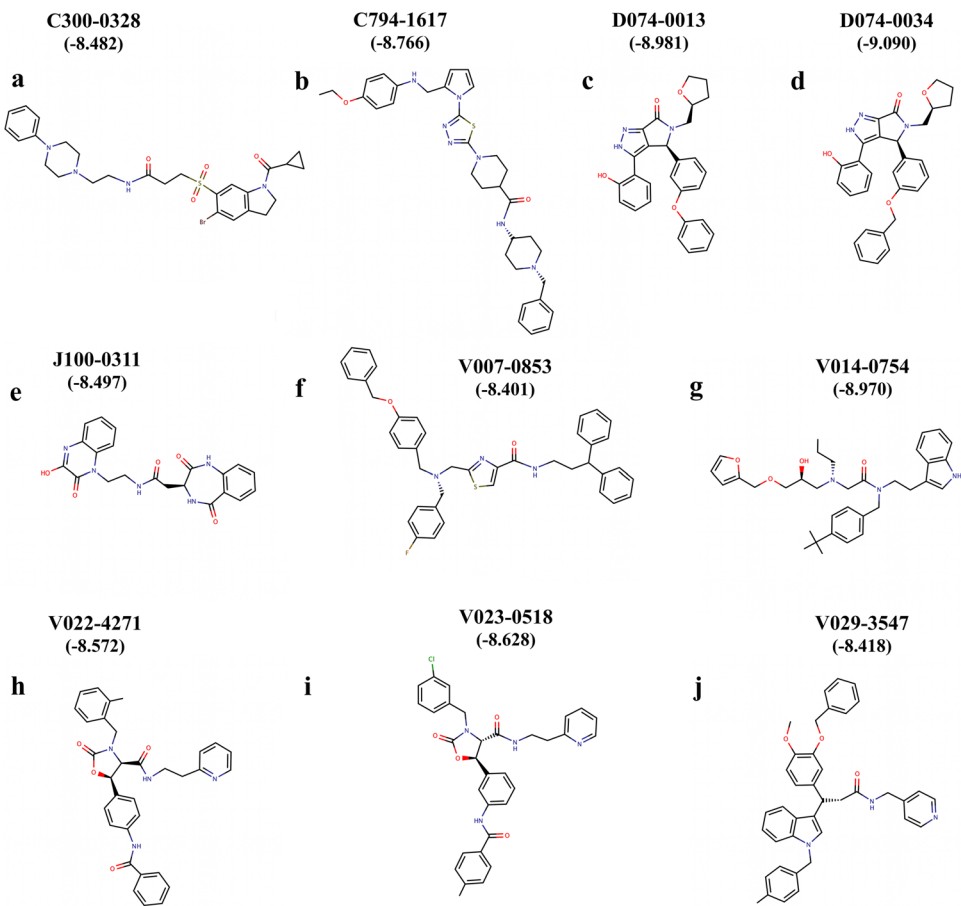

**Fig. 2 | Virtual screening studies provide ten molecules for further applications.** Compounds shown in (**a**–**d**) and (**f**–**j**) are from the GPCR-Targeted library. **e** is the only compound from the Peptidomimetic library. The Glide docking score (kcal/mol) of each molecule is given below the compound name. Black: carbon atoms, red: oxygen atoms, blue: nitrogen atoms, green: chlorine atoms, gray: sulfur atoms, dark brown: bromine atoms, and light brown: fluorine atoms. Source data are provided as a Source Data file.

bonds were dominant between the D074-0034 and the receptor (Supplementary Fig. 4d). In contrast to the D074-0013, D074-0034 interacted with the Gln271 thoroughly, although it still preserved the interactions with Asn188. Additionally, it also has π-π stacking interactions with Phe291.

The peptidomimetic ligand J100-0311 was noteworthy for its number of continuous interactions with the binding pocket residues (Supplementary Fig. 4e). Hydrogen bonds with ECL-2 residues Ile179, Asn183, Gly184, and Ser186 were preserved robustly, indicating the importance of ECL-2 in GPCR activation[47]. Gln278 was also significant in sustaining its contacts with the ligand.

Analysis of the V007-0853:AlstR-C complex demonstrated that interactions primarily involved hydrophobic contacts with the Trp191 and Phe291, which were observable throughout the simulations (Supplementary Fig. 4f). Similarly, Trp191 was the most interacted residue in the V014-1754 compound (Supplementary Fig. 4g). However, the ligand had no other significant noncovalent interactions except Leu274 with the receptor that persisted through the simulations.

Similar contacts were encountered in V022-4271:AlstR-C simulations, such as π-π stacking interactions with Phe291, hydrogen bonding interactions with Trp191, and polar contacts with Asn183 and Asn188 residues (Supplementary Fig. 4h). Moreover, ion pair interactions were distinguished with Glu193 and Glu281 residues. Subsequently, the same Glu281 ionic interaction was observed with the V023-0518 (Supplementary Fig. 4i). Polar and ionic interactions were also found to occur in the binding site, especially for Thr280,

Ser186, and Asn188 alongside the hydrophobic contacts. Lastly, prominent interactions with residues such as Asn188 and Ser186 were recorded in V029-3547:AlstR-C simulations beside the Trp191 (Fig. 3, Supplementary Fig. 4j).

## Binding free energies of receptor-ligand complexes represent the potencies of the hit compounds

MD simulations were further analyzed to assess binding free energies (ΔG) of protein-ligand complexes, and 100 out of 1000 saved trajectory frames (protein-ligand complex structures) during the simulations were subjected to molecular mechanics generalized Born surface area (MM/GBSA) analysis. The J100-0311 molecule was the ligand with the lowest average binding free energy to the AlstR-C receptor, with an average ΔG value of −101.37 ± 10.41 kcal/mol. The compound V007-0853 was the second tightest binding ligand with average ΔG score of −100.98 ± 10.61 kcal/mol. The hit compounds D074-0013 (−93.58 ± 9.87 kcal/mol), V029-3547 (−92.81 ± 10.54 kcal/mol), and D074-0034 (−91.78 ± 8.47 kcal/mol) had similar average ΔG values. The highest average ΔG value (i.e., weakest ligand) was V014-0754 with −57.13 ± 10.70 kcal/mol, which also showed the lowest structural stability in simulations (Supplementary Fig. 5a).

Ligand efficiency is an expression of a ligand's binding energy per nonhydrogen atoms to its binding partners, such as a receptor. Ligand efficiency normalizes the binding affinity of compounds with respect to molecular size. The molecule with the top ligand efficiency is found to be J100-0311, which has the lowest average MM/

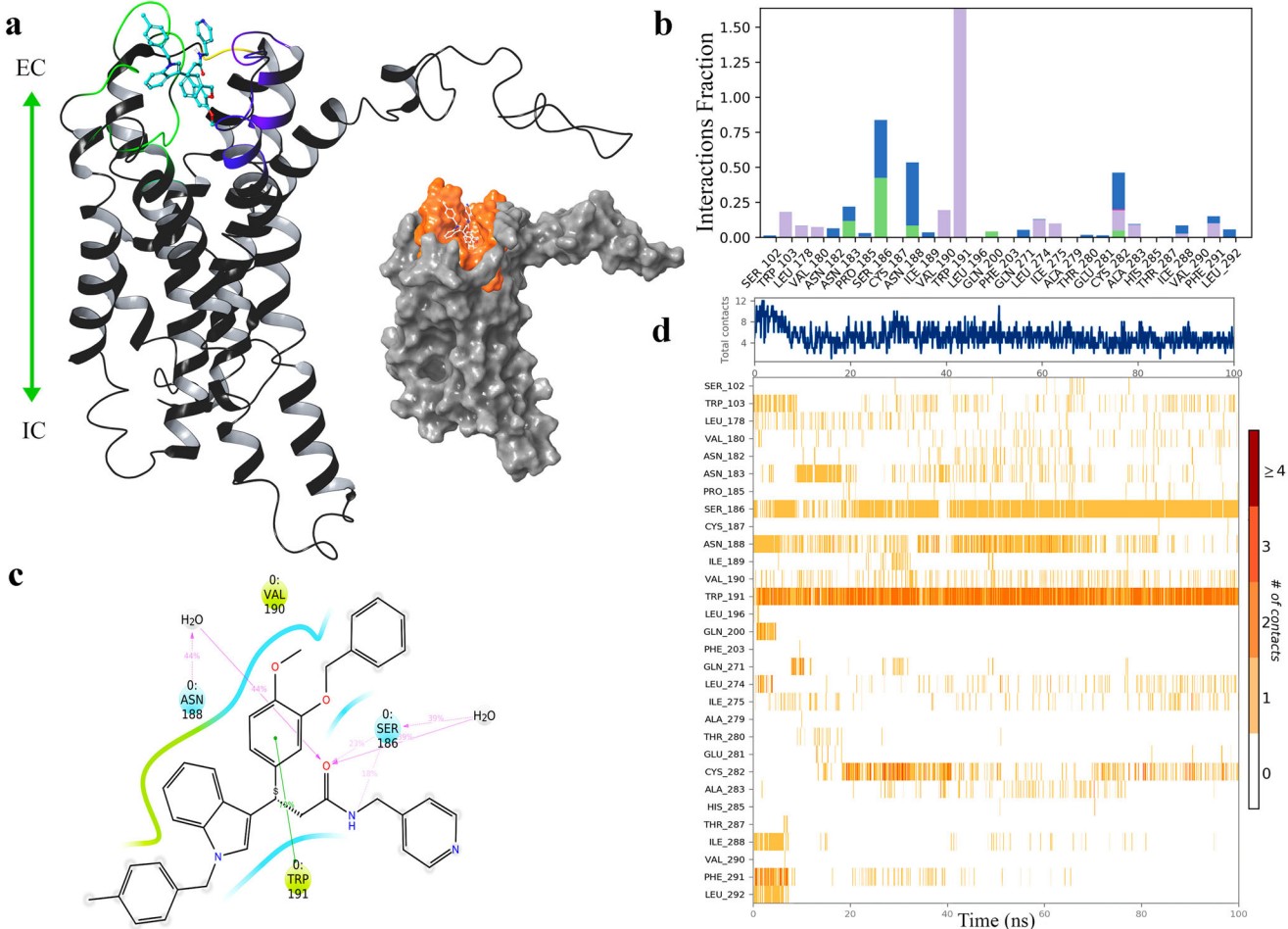

**Fig. 3 | MD simulations show robust contacts between AlstR-C and V029-3547.**
**a** Ribbon and surface representation of V029-3547:AlstR-C complex. V029-3547 is at the orthosteric pocket, located at the receptor's extracellular site. Ligand is colored blue and represented in ball and stick format. Loop regions in contact with the compound are colored yellow (ECL1), green (ECL2), and purple (ECL3). The orange-colored surface shows the binding pocket of AlstR-C. **b** Protein-ligand interactions plot shows the contacts throughout the simulations. Stacked bar charts are normalized according to simulation time. Purple: hydrophobic interactions, green: hydrogen bonds, blue: water bridges, and pink: ionic interactions.

**c** Detailed representation of interactions of active site residues and ligand atoms. Interactions that occur more than 15.0% of the simulation time, are shown. Lime color shows hydrophobic contacts, and blue illustrates the water bridges. Green is an indicator of π-π stacking interactions. **d** Illustration of the interactions and contacts throughout the simulation time. The top panel depicts the total number of contacts, while the bottom panel displays which residues interact with the ligand in each trajectory frame. A deeper shade of orange denotes specific residues that make more than one contact with the ligand. Source data are provided as a Source Data file.

GBSA, with −3.02 ± 0.32 kcal/mol per nonhydrogen atom (Supplementary Fig. 5b).

## TGF-α shedding and Glo-sensor assay confirms the activities of identified hit compounds

Based on average MM/GBSA binding free energy scores of ligands, the top five compounds were ordered for in vitro analysis, but only four were available (V007-0853 was out of stock). These four molecules (D074-0013, D074-0034, J100-0311, and V029-3547) were screened against *T. pityocampa* AlstR-C in cells overexpressing the receptor. TGF-α shedding assay was used on HEK293 cells to determine the pharmacological properties of the molecules (Fig. 4). The natural peptide of the receptor AST-C was used as a positive control to compare the binding affinity of the candidate ligands.

As the receptor's natural ligand, the AST-C peptide showed the highest affinity since it's the receptor's natural ligand with an $EC_{50}$ value of 0.623 nM (Fig. 4a). Notably, the hit compound D074-0013 also demonstrated significant agonistic activity with an $EC_{50}$ value of 1.421 μM (Fig. 4c). However, its derivative, D074-0034, displayed approximately a 3-fold decrease in binding affinity at the target

protein compared to D074-0013, registering an $EC_{50}$ value of 4.700 μM (Fig. 4e). This outcome suggests that the addition of a $CH_2$ group between two phenyl groups might negatively impact the proper interactions of D074-0034 at the binding site. The calculated average binding free energy results for these two analogs align well with the experimental findings, where the average MM/GBSA scores for D074-0013 and D074-0034 were computed as −93.583 and −91.781 kcal/mol, respectively. However, despite the similarity in the measured docking scores of these analogs (−8.981 for D074-0013 and −9.090 kcal/mol for D074-0034), this result hints that relatively static calculations in docking simulations may not be sufficient to discern subtle differences in binding. Ligand J100-0311, the only peptidomimetic compound, displayed an $EC_{50}$ value of 5.662 μM agonist activity (Fig. 4g). Another identified hit compound, V029-3547, represented the lowest binding affinity among the tested ligands at the AlstR-C ($EC_{50}$ of 8.174 μM) (Fig. 4i). However, it must be highlighted that the $EC_{50}$ values show no dramatic differences between the binding affinities of the tested hit molecules. The $EC_{50}$ values of tested four hit compounds were between 1.421 and 8.174 μM.

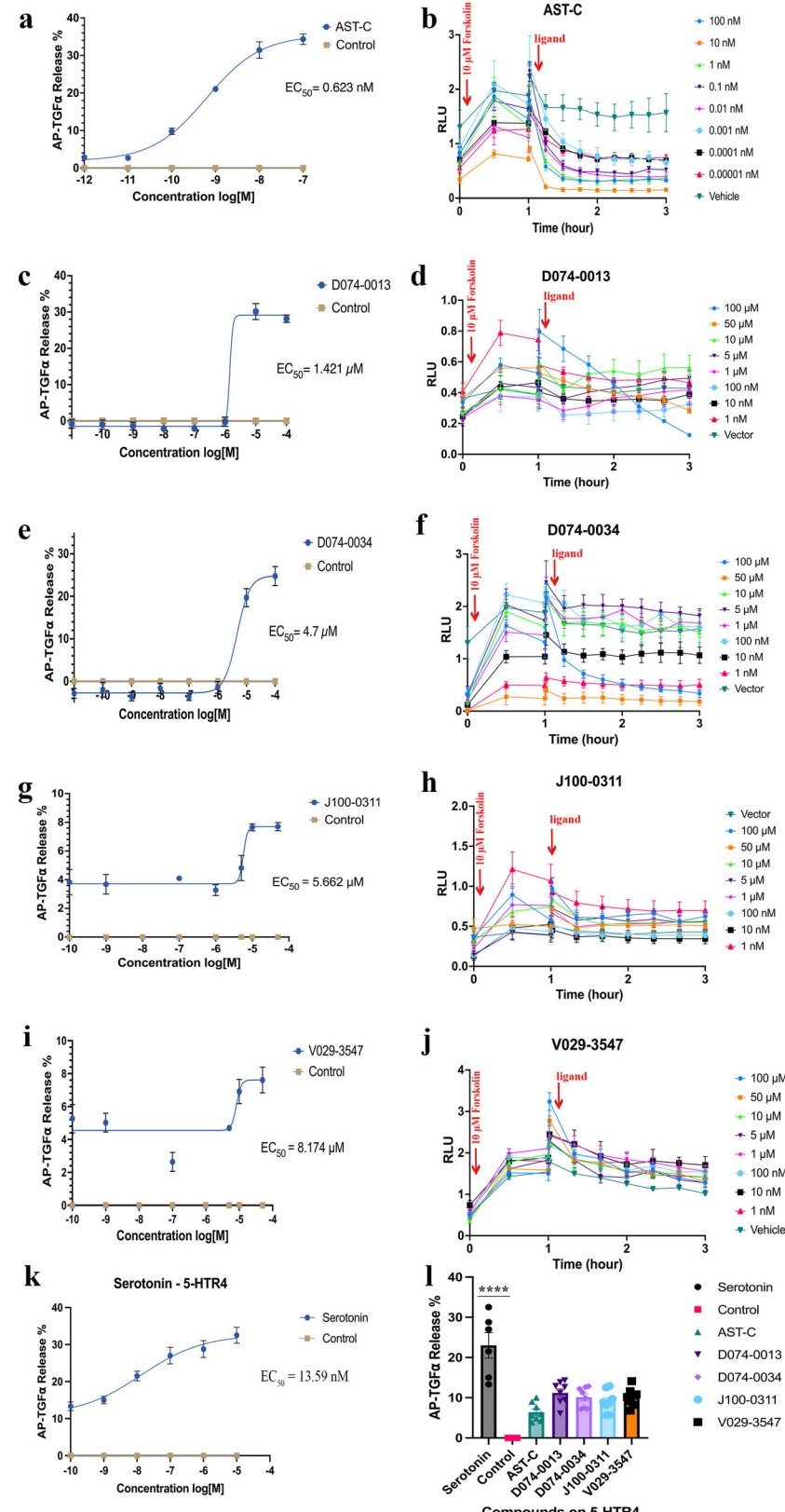

## Glo-sensor assay confirms downstream activation of Gα$_i$ pathway in hit compounds

The ability of the hit compounds to activate the Gα$_i$ pathway was evaluated using a commercial cAMP assay (GloSensor, Promega). Prior to ligand treatment, cells were stimulated with 10 μM forskolin to elevate intracellular cAMP levels, and the ability of each ligand and control to downregulate cAMP was evaluated after 1 h. The native AST-C ligand exhibited the most significant impact (**** $P < 0.0001$) (Fig. 4b), while V029-3547 was the hit molecule with the greatest ability to modulate cAMP compared to the control ($F = 14.62$ *** $P < 0.05$) (Fig. 4j). As an additional control, the ability of the hit compounds to activate a different receptor (5-HTR4) was also assessed using an AP-

**Fig. 4 | TGF-α shedding assay reveals agonist properties of the predicted ligands. a–j** AP-TGFα release responses of ligand-treated AlstR-C on the left. (Statistical significance was assessed by a one-tailed Student's $t$-test for each molecule $P < 0.05$ ($n = 4$)). Real-time molecule treatment effects on cAMP production on the right. Concentration-dependent decreases in RLUs were observed at all treatments (Two-way ANOVA, $P < 0.05$, Dunnett's Test, D074-0013: Adjusted $P < 0.0001$, D074-0034: Adjusted $P$ value 0.0042, J100-0311: Adjusted $P < 0.0001$, and V029-3547

Adjusted $P$ value 0.0403). Error bars, SEM for four biologically independent replicates from one experiment ($n = 4$). **k** AP-TGFα release after Serotonin treatment on 5-HTR4 expressing cells. **l** The hit molecules were tested on the 5-HTR4 receptor. Molecules did not show a significant difference among each other compared to Serotonin, the natural ligand of the 5-HTR4 (One-way ANOVA, $P < 000.1$, $F = 30.97$, $n = 4$ biologically independent replicates). Source data are provided as a Source Data file.

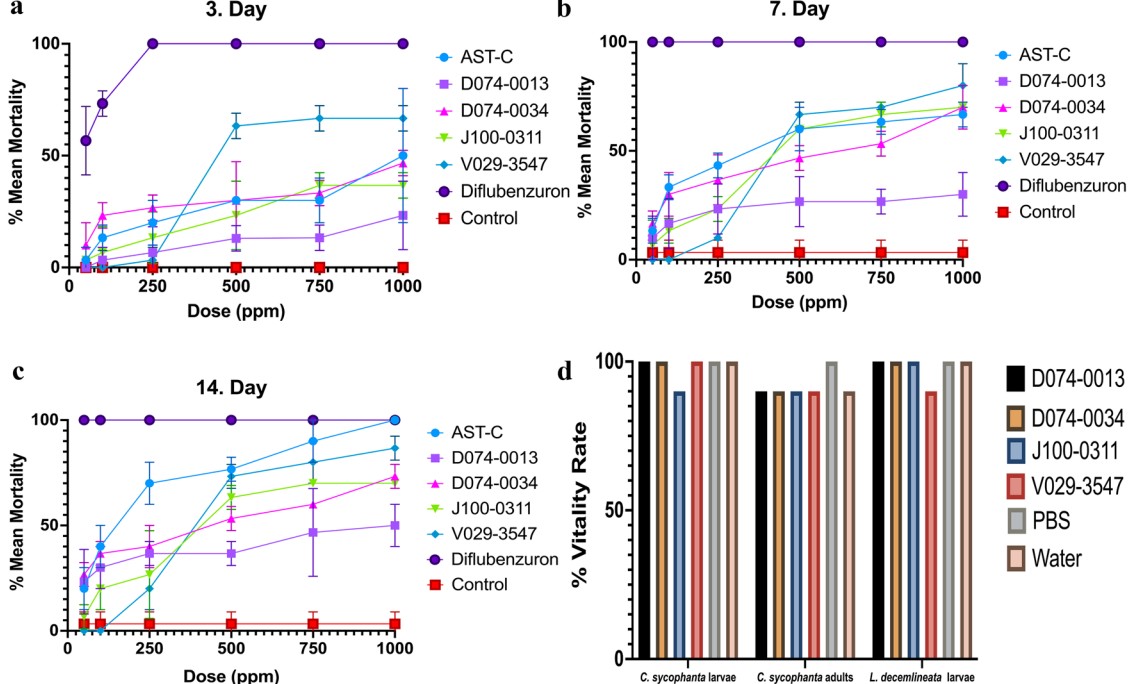

**Fig. 5 | The in vivo experiments validate insecticidal properties of candidate molecules and specificity towards *T. pityocampa* AlstR-C. a–c** Results of one-time application of different doses of compounds on *T. pityocampa* larvae on days 3, 7, and 14. Each dose application has three independent replicas ($n = 10$), and error bars represent the standard deviation (SD). **d** Stack bar chart illustrates the side

effects of 1000 ppm of molecules on *C. sychophanta* larvae ($n = 50$), adults ($n = 50$), and *L. decemlineata* larvae ($n = 30$). Mortality rates of applied molecules are not significantly different from negative controls (Two-way ANOVA and Tukey test, $P > 0.05$). Data are representative of three independent experiments. Source data are provided as a Source Data file.

TGFα assay (Fig. 4k). TGFα shedding induced by each molecule was comparable to AST-C and statistically insignificant compared to serotonin, suggesting that the pesticides are specific to AlstR-C (Fig. 4l).

## Hit compounds are lethal to *T. pityocampa* in vivo at doses comparable to the native ligand

Following the confirmation of in vitro efficiency, the hit compounds and natural ligand AST-C were tested in *T. pityocampa* larvae in vivo. These compounds were evaluated alongside with the pesticide Dimilin 25 WP (Diflubenzuron 25%), a commercially used larvicide against *T. pityocampa* and other pests. A dose and day-dependent effects of the hit compounds and controls were determined on second instar larvae. The lethality results of active compounds on the 3rd, 7th, and 14th days are displayed in Fig. 5. In the negative control group, the mean mortality was 0% on day 3 and 3.3% on days 7 and 14. In the positive control groups, 56.7% and 73.3% mortality were observed at doses as low as 50 and 100 ppm on the 3rd day, with 100% mortality for all other doses and days.

Mean mortality values remained below 50% after 3 days of administration for all compounds except V029-3547, which showed the highest efficacy after Dimilin with >63.3% mortality at >500 ppm ($F = 40.1$, $p < 0.01$) (Fig. 5a). At >750 ppm, more than 50% mortality was observed after 7 days for all hit compounds except D074-0013 ($F = 89.8$, $p < 0.01$) (Fig. 5b). At the end of day 7, the lowest mortality at

1000 ppm was recorded at 27% with the D074-0013 agonist ($F = 54.4$, $p < 0.01$).

Mean mortality rates for 14 days post-administration reached the highest percentage for all ligands. According to the results of the Tukey test, all compounds applied at doses of 500 ppm and above caused a significant decrease in the number of live larvae compared to the control (Fig. 5c). While the AST-C ligand showed the highest larvicidal activity with 100% mortality, the second highest effect was observed at 83.7% with V029-3547 ($F = 121.1$, $p < 0.01$). However, even at the highest dose, D074-0013 exhibited a mortality rate below 50%.

Toxicity data on the day 14 after the application were evaluated with the Probit Analysis method and LC$_{50-90}$ data were obtained and presented in Fig. 6. LC$_{50}$ values of AST-C, D074-0034, J100-0311, and V029-3547 hit compounds are 152, 443, 411, and 406 ppm, respectively. In contrast, LC$_{50}$ value was >1000 ppm for D074-0013 (Table 1). Overall, the AST-C ligand had the highest effect compared to other compounds.

In addition, the ability of the identified hit compounds to alter larval body allometry was tested through measuring total body lengths and head capsule widths on the 14th day of treatment (Fig. 7). Statistically significant decreases in both parameters were observed for AST-C and all four of the identified hit ligands, albeit the effect of D074-0013 was small compared to other molecules ($p = 0.0004$). In contrast,

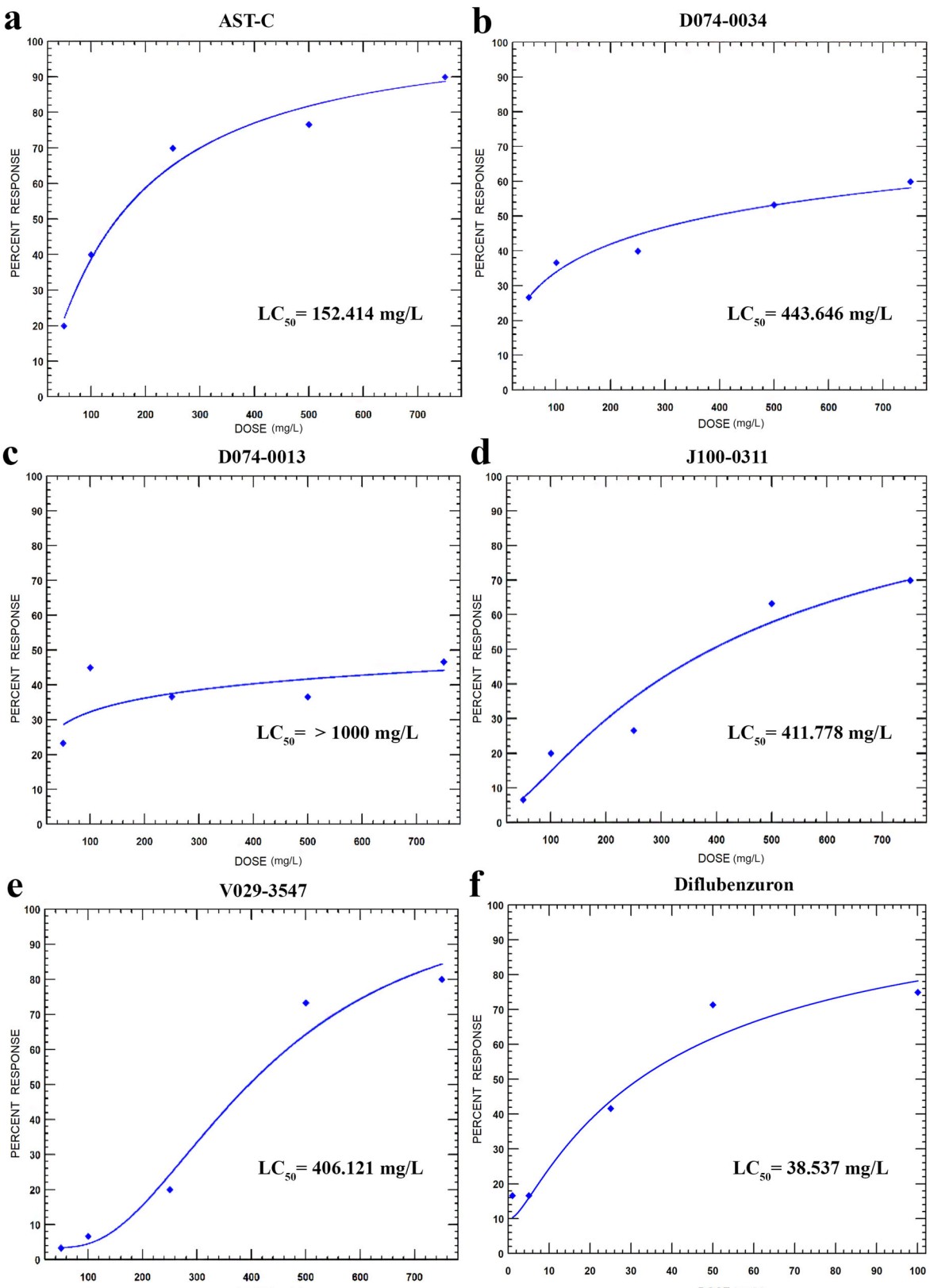

**Fig. 6 | Probit analysis revealed the lethality concentrations of pesticide candidates compared to the natural ligand. a** Natural ligand of the AlstR-C receptor of *T. pityocampa* (**b**–**e**) Pesticide candidates. **f** Lethal dose of diflubenzuron on *L. decemlineata* as a positive control. Lethal concentration values of $LC_{50-90}$ with 95% CI ($P > 0.05$). Slopes were obtained over 14. day mortality values ($n = 180$). Source data are provided as a Source Data file.

**Table 1 | Lethality concentrations of compounds on *T. pityocampa* larvae in vivo**

| Compound | N | LC$_{50}$ (95% CI) (mg/L) | LC$_{90}$ (95% CI) (mg/L) | Slope ± SE | $\chi^2$ (df) |
|---|---|---|---|---|---|
| AST-C | 180 | 152.414 (101.993–209.528) | 782.851 (512.785–1616.980) | 1.803 ± 0.299 | 0.974 (3) |
| D074-0013 | 180 | >1000 | - | 0.377 ± 0.266 | 2.280 (3) |
| D074-0034 | 180 | 443.646 (207.273–2759.441) | 24448.685 (3423.838–9.734 × 10$^8$) | 0.736 ± 0.262 | 2.143 (3) |
| J100-0311 | 180 | 411.778 (282.472–600.956) | 6740.368 (2743.061–6.275 × 10$^4$) | 1.916 ± 0.429 | 0.666 (3) |
| V029-3547 | 180 | 406.121 (305.704–492.464) | 900.815 (696.503–1634.450) | 3.704 ± 0.871 | 2.053 (3) |

**Fig. 7 | Effects of the four identified hit compounds, the native peptide AST-C, untreated application medium (PBS with 0.1% BSA), and a non-AST-C targeting control ligand (7119812967) on *T. pityocampa* larval length and head capsule width.** Left: Changes in larva body length are demonstrated following the 14th day after being treated with pesticide candidates, AST-C, and control ligand 7119812967. *P* values are given above the bars (*n* = 20). Right: Effects of treatment with hit compounds, native peptide, and 7119812967 on head capsule width. *P* values are given above the bars (*n* = 20). Each *n* corresponds to an individual larva. Numbers above bars denote *P* values as determined by one-way ANOVA. Tukey's test was used for multiple comparisons. Error bars are presented in ±SEM. Source data are provided as a Source Data file.

no significant reduction in body length was observed in negative control compound 711981296, a compound from Otava peptidomimetic library, that our group had previously identified in silico as a potential AlstR-C agonist while we observed no in vitro effect on the receptor in a GPCR activity assay[48].

The effects of native ligands, naturally occurring molecules in the body, are often higher than those of chemical ligands due to their specific interactions with biological targets. Native ligands have evolved to bind and interact with their target molecules in a highly specific and efficient manner, leading to stronger and more pronounced effects. Additionally, native peptide ligands carry certain challenges and limitations as therapeutic targets such as oral bioavailability, short half-life, poor membrane permeability, structural stability, synthesis and manufacturing complexity. Despite these challenges, advancements in peptide engineering, formulation, and delivery methods are addressing some of these limitations. Modified or engineered peptides, as well as innovative delivery systems, are being developed to enhance stability, bioavailability, and overall efficacy of peptide-based therapeutics. Examples include the development of peptide mimetics, conjugation with other molecules to improve pharmacokinetics, and the use of nanoparticle delivery systems.

### Side effects of the identified hits on other insects were minimal

The side effects of D074-0034, J100-0311, V029-3547, and D074-0013 compounds were tested on the first-instar larvae and adult stages of the predatory insect *Calosoma sycophanta* and the second-instar larva of the ten-lined potato beetle (*Leptinotarsa decemlineata*) by topical application. Mortality rates were evaluated on the 3rd, 7th, and 14th days after the application. Insect deaths remained below 10% and showed similar ratios to the negative control (Fig. 5d). The side effect examinations showed that identified hit compounds in this study are specific for the *T. pityocampa* AlstR-C receptor. They did not have an insecticidal effect on larvae and adults of *C. sycophanta* and potato beetle larvae. However, potato beetle larvae displayed high susceptibility to diflubenzuron with $LC_{50}$ value of 38 ppm after 3 days of application.

## Discussion

Recent studies have shown that many neuropeptides are associated with GPCRs in insects, including AST-C[49,50]. Although allatostatins exist in many insect species, not all allatostatin types are expressed in every order[51,52]. AlstR-C was identified in many insects such as *C. morosus*, *D. melanogaster*, and recently in *T. pityocampa* by our group[44,48,53]. JH and allatostatin dynamics are essential contributors to insect development regulation, and are extensively studied in this context. Despite the structural conservation of AlstRs, their expression profiles and functions are versatile within and between insect orders[52,54]. Consequently, understanding the mechanism behind the AlstRs and JH dynamics is a compelling concept for understanding the developmental and behavioral aspects of insects and developing efficient control agents. Considering the inadequate pest control systems and raising concerns about pest resistance and environmental hazards, next-generation pesticides are becoming increasingly necessary[55].

In a previous study of our group[44], the AlstR-C target and its natural ligand AST-C of *T. pityocampa* were investigated using in silico and in vitro biological assays. Despite the widespread ecological damage and potential health hazards caused by this insect, its control is currently limited to pheromone traps, manual removal of nests and eggs, and use of broad-spectrum larvicidal chemicals such as diflubenzuron. In addition, increasing global temperatures have allowed the pine processionary to colonize new habitats and potentially act as an invasive species[56]. Our study provides a series of promising candidate pesticides for the effective and selective control of this pest, and the combination of computational, in vitro and in vivo methods outlined here may serve as a guideline for the design of other pest-specific control agents.

The effects of native ligands are often higher than those of chemical ligands due to their specific interactions with biological targets. However, the bioavailability of peptides is a crucial factor when considering their effectiveness as a pesticide. Peptides can be susceptible to degradation by environmental factors, such as sunlight, heat, and enzymes[57,58]. This degradation can reduce the amount of active peptide reaching the target pests, affecting bioavailability. Formulating peptides into a stable and effective product is essential for enhancing bioavailability. Proper formulation can protect peptides from degradation and improve their delivery to the target organisms. The method of applying peptides to the target area can influence their bioavailability. Factors such as spraying, injection, or soil application can affect how well peptides reach and interact with the pests. For a pesticide to be effective, it must be taken up by the target pests. The ability of peptides to penetrate the pest's cuticle or membranes and reach their internal systems can impact bioavailability. The persistence of peptides in the environment also plays a role in their bioavailability. If peptides break down too quickly, they may not remain active long enough to exert their intended effects. Considering all these factors adds to the complexity of utilizing peptides as pesticides.

As a peptide, AST-C exhibits a comparatively short shelf life and undergoes rapid degradation in the environment[57,58]. Furthermore, administering it orally is impractical due to digestion concerns[58]. Although attempts have been made to enhance the stability of allatostatins through methods like D-isomer substitution, small organic compound ligands such as peptidomimetics present a more stable alternative with the potential for cost-effective mass production efforts[59-61].

In search of a next-generation pesticide for *T. pityocampa*, virtual screening and MD simulations were applied to AlstR-C and based on the docking scores, ten molecules were used in 100 ns MD simulations with three replicas. Simulations were evaluated based on different criteria for these molecules. RMSD changes of AlstR-C:Hit compound complexes were calculated for $C_\alpha$ of residues and translational and rotational movements of the molecules in the binding site. This analysis yielded varying results for each complex; $C_\alpha$ plots demonstrated RMSD values ranging from 2 to 4 Å, and the lowest RMSD was observed in AlstR-C:V022-4271 complex with an average value of 2.095 Å. However, it was drastically different for Lig-fit-Protein RMSDs. Some of the compounds reached equilibrium later and underwent conformational changes compared to the reference position of the ligand. Further assessment of the ligand binding to the receptor was accomplished via MM/GBSA binding free energy analysis. Four compounds were chosen for cell-based and in vivo assays considering the average binding free energies, contacts with the residues, and stability of these interactions at the binding cavity of the target protein.

Our TGF-α shedding assay suggests that the four identified hit compounds in this study all had similar $EC_{50}$ values (between 1.421 and 8.174 μM), with D074-0013 exhibiting the strongest effect with an $EC_{50}$ value of 1.421 μM. In contrast, V029-3547 consistently had the strongest larvicidal effect in in vivo studies. While the shedding assay is a well-established indicator of GPCR activation, especially for the $G\alpha_q$ and $G\alpha_{12/13}$ pathways, its efficiency is limited for the $G\alpha_i$-dominant signaling profile of AlstR-C[44]. In addition, the transition from in vitro to in vivo models is invariably complicated by the complex network of feedback mechanisms that maintain homeostasis in live animals, with absorption, distribution, metabolism, and excretion (ADME) mechanisms having a particularly significant impact on in vivo drug effectiveness. It should also be noted that, while our method of applying 5 μL pesticide onto the larval cervix follows literature precedent (see e.g., 1 μL applied to dorsal surface in refs. 62,63, 2 μL in refs. 64,65, 5 μL in refs. 66,67 and the instars are large enough to absorb the entire volume (Supplementary Fig. 6), it is feasible that part of this volume fails to contact the cuticle due to protective hairs or is transferred the substrate following application. As such, some discrepancy between in vitro and in vivo studies are unavoidable, and more comprehensive experiments involving metabolic assays should ideally be undertaken to better establish the effectiveness of the hit molecules.

The delicate balance between JH production and degradation is mandatory for proper insect growth and metamorphosis[20]. Activation of AlstR-C is a core element of this process, and its dysregulation may be the primary reason for the mortality observed in hit compound-treated larvae. Failure to molt and deviations in body size and allometry are primary consequences of JH dysregulation[68]. Consistent with this view, both AST-C and hit compound treatments were associated with statistically significant decreases in body length and head capsule width (Fig. 7). In contrast, a ligand from the Otava Chemicals Peptidomimetic Library (which we had previously identified as a potential AlstR-C hit in silico in a previous study, but failed to yield in vitro activity) was used as a negative control and did not yield a statistically significant difference compared to PBS control[48]. The higher mortality values at the end of the 14th day may be due to the decrease in feeding by preventing the foregut motility of the larvae of the pine processionary moth. In addition, a previous

study reported that the receptor for *Manduca sexta* allatostatin (Manse-AS) inhibits foregut contraction in lepidopteran larvae, suppressing feeding and potentially contributing to larval mortality[69]. The researchers observed that fifth instar *Lacanobia oleracea* larvae injected with Manse-AS between the head and prothorax had a decline in growth, decreased feeding, and increased mortality compared to the control. Correspondingly, when *Acyrthosiphon pisum* and *Myzus persicae* were fed with artificial food containing Manse-AS, it caused growth decline, decreased fecundity, and significant death due to suppression of nutrition[58].

AlstRs are known to regulate various other physiological processes, including hormone production and release. By binding to this receptor, the agonist may disrupt the normal signaling pathways involved in JH regulation, leading to substantial changes in hormone levels or behavior. Similarly, there was a decrease in feeding and defecation, deterioration in netting behavior, developmental retardation, and the absence of the characteristic "pine processionary larvae marching" behavior in the *T. pityocampa* larvae treated with the hit compounds. In contrast to their significant effects on *T. pityocampa* survival, allometry and behavior, these compounds were associated with low mortality rates on coleopteran larvae and adults. Low mortality rates in other adult insects and larvae suggests that the identified hit ligands are specific at least to Lepidoptera, allowing the design of safer and more selective next-generation pesticides. Consequently, the identified hit compounds found to be effective in this study can be used and further developed as next-generation insecticides in AlstR-C-targeted *T. pityocampa* control.

In the current study, we utilized the advances in rational drug design and cell-based and in vivo approaches to obtain target-specific and safer pesticides for *T. pityocampa* AlstR-C. Overall, our study is an important initial advance toward an insect GPCR-targeted next-generation pesticide design. Our approach may apply to other invertebrate GPCRs involved in vital metabolic pathways.

## Methods

### Ligand preparation
Peptidomimetic[70] and GPCR-Targeted[71] libraries from ChemDiv were prepared with LigPrep module of Maestro molecular modeling package using OPLS3e[72] force field. Ionization was applied to generate possible protonation states at pH 7.0 using Epik. The specified chiralities were retained during the ligand preparations. After ligand preparation, 61,518 structures from the Peptidomimetic library (initial number: 36,711) and 64,515 structures from the GPCR-Targeted library (initial number: 29,185) were produced.

### Protein preparation
The Protein Preparation module of the Maestro molecular modeling package[73] was used to prepare the receptor. The bond orders were assigned, hydrogens were added, and disulfide bonds were created. Subsequently, protein refinement and minimization were performed. Hydrogen bond assignments were optimized at pH 7.4 using the PROPKA (Schrödinger, LLC, New York, NY, 2018). Restrained minimization was performed with the OPLS3e force field.

### Virtual screening
Ligand docking was applied to investigate the interactions of ligand libraries on the orthosteric pocket. Grid was generated with the Receptor Grid Generation module, Glide[74], based on the natural ligand's position in the orthosteric pocket. The virtual screening was executed with the prepared ligand libraries using Glide[74]. Peptidomimetic library (61,518 molecules) and GPCR-Targeted library (64,515 molecules) were docked separately, but for the hit molecules, docking scores were evaluated together (126,033 molecules). The Standard Precision (SP) module and flexible ligand sampling were used. Epik state penalties were added to the docking scores. Intramolecular

hydrogen bonds were rewarded, and the planarity of conjugated pi groups was enhanced. Post-docking minimizations were performed, and five poses were included per ligand. The molecules to be subjected to MD simulations were chosen according to docking scores, pharmacophore units, and receptor contact residues.

### Benchmarking virtual screening
To benchmark the docking studies, we utilized the GPCR Decoy Database provided by the Cavasotto Laboratory[75]. The decoys were selected according to physical descriptors for the four active molecules using DecoyFinder[76] with the default settings. 36 decoys were generated for each of them, yielding 144 molecules in total. The decoy library underwent identical processes and parameters as the active ligands. Consequently, ligand preparation yielded 431 molecules. The generated decoy library was docked to the receptor (see Supplementary Fig. 1), and the 10 best-performing compounds were subjected to MD simulations and MM/GBSA binding free energy analysis. During the selection process, Gscore and Glide Emodel energies were considered for the multiple structures of the same molecules.

### System preparation
Desmond[77] System Builder module was used for system setup for MD simulations. Membrane orientation was set up according to PDB:6DDE from the OPM database (https://opm.phar.umich.edu/). Next, the target protein was embedded in a POPC (1-palmitoyl-2-oleoyl-sn-glycerol-3-phosphocholine) lipid bilayer, and the TIP3P water model was selected. The structure was neutralized by adding counter ions, and finally, 0.15 M NaCl was added to the system.

### MD simulations
OPLS3e[72] force field was used, and the equilibration step was performed using the default settings. The simulation temperature was set as 310 K, and the pressure was 1.01325 bar. The Nose−Hoover thermostat[78] and the Martyna−Tobias−Klein barostat[79] methods were applied to the system. The particle mesh Ewald[80] method was applied to calculate the long-range electrostatic interactions. A cut-off radius of 9.0 Å was used for both van der Waals and Coulombic interactions, and the time step was assigned as 2.0 fs. An NPγT ensemble was used during the production step of MD simulations with a surface tension of 4000 bar/Å. 100 ns MD simulations were performed with three independent replica simulations, and concatenated trajectories were used in analyses. MD simulation trajectories were analyzed with the Simulation Interaction Diagram module.

### Molecular mechanics with generalized Born and surface area solvation (MM/GBSA) binding free energy analysis
Out of the total 1000 trajectory frames of each MD simulation, 100 trajectory frames were subjected to MM/GBSA calculation. The VSGB 2.0 solvation model in the Prime, Schrödinger, LLC, New York, NY, 2018, was utilized[81]. Each compound's average MM/GBSA binding free energies to AlstR-C were calculated for these 100 frames. Average MM/GBSA analyses were repeated for each replica simulation.

### Bioassay
All the finally selected compounds with at least 95% purity were ordered from ChemDiv (https://www.chemdiv.com/). The purity accuracy is confirmed by ${}^1$H NMR and LC (UV)/ MS spectra.

### Cell culture and transfection
HEK293 cells were maintained in DMEM (PAN Biotech) supplemented with 10% FBS (Gibco) and 1% Penicillin-Streptomycin (Gibco) at 37 °C in a 5% $CO_2$ incubator. Cells were seeded in 6-well plates before transfection, and Lipofectamine™ 3000 Transfection Reagent (Thermo Fisher Scientific) was used for transfection in compliance with the manufacturer's instructions.

## TGF-α shedding assay

HEK293 cells were seeded in DMEM 10% FBS without P/S to 6-well plates 1 day before the transfection. Following overnight incubation at 37 °C, the cells were cotransfected with plasmids of the desired receptor (100 ng)[44,48] or pcDNA3.1, AP-TGFα (250 ng), and Gαq/i1 (10 ng) using Lipofectamine™ 3000 Transfection Reagent and incubated 24 h at 37 °C 5% $CO_2$ incubator. Plasmids used for the TGF-α shedding assay were kindly gifted from Prof. Asuka Inoue[82,83]. TGF-α Shedding Assay was used to measure the activity of studied compounds[83]. Cell harvesting was accomplished with TrypLE™ Express Enzyme (Gibco) since it was previously noted that trypsinization results in a higher background signal[83]. After reseeding the cells in a 96-well plate, 10x concentrations of the compounds prepared in HBSS were applied for stimulation. For compounds not readily soluble in water, a concentration of 0.01 % (w/v) bovine serum albumin (BSA) in HBSS was used for dilution[82]. Each treatment was performed with four replicas. Alkaline phosphatase activity was measured before and after 1-h and 2-h incubation at room temperature, and the absorbance of the plates at 405 nm was measured. AP-TGF-α release signals were fitted to a four-parameter sigmoidal dose-response curve, and $EC_{50}$ values were calculated using GraphPad Prism 9.5.

## GloSensor cAMP Assay

HEK293 cells were seeded 30,000 cells per well in DMEM 10% FBS without P/S to Greiner 655094 96-well plates 1 day before the transfection. Following overnight incubation at 37 °C, the cells were cotransfected with plasmids of the desired receptor along with GloSensor™ cAMP plasmid using Lipofectamine™ 3000 Transfection Reagent and incubated 24 h at 37 °C 5% $CO_2$ incubator. Plasmids used in cAMP assay were kindly gifted from Prof. Meliha Karsak, University of Hamburg. On the experiment day, media was aspirated, and 80 µL beetle luciferin 2 mM/Leibovitz's L-15 Medium, no phenol red 1% FBS solutions were added to each well. The baseline development was measured on a Thermo Scientific Fluoroskan Ascent FL Microplate Fluorometer and Luminometer plate reader following the cell stimulation with 10 µL 10 µM forskolin, approximately 1 h. Consequently, to signal stabilization, cells were stimulated with vehicle control (Leibovitz-beetle luciferin) or different ligand concentrations. Luminescence was measured at 1 read per well every 2 min, scaling factor was 10. Data were analyzed by calculating the area under the curve (AUC) over the ~2-h measurement period. Statistical significance was measured with ANOVA, and multiple comparisons were done with Dunnett's Test. All calculations were done using GraphPad Prism 10.

## In vivo studies

Pine processionary moth 2nd instar larvae of *Thaumetopoea pityocampa* for toxicity tests, adult, and 1st instar larvae of predator *Calosoma sycophanta*, 2nd instar larvae of potato beetle *Leptinotarsa decemlineata* were used for side effect assays.

*T. pityocampa* larvae were collected from red pine trees, *Pinus brutia*, in Nallıhan district of Ankara, Türkiye, in 2021. Larvae were cultured in 60 (±10) % humidity, 27 (±2) °C, and 16:8 light: dark photoperiod and were fed with red pine needles.

*C. sycophanta* adults and *T. pityocampa* nests were collected from Korkuteli forests in Antalya 2022. *T. pityocampa* nests were stored at +4 °C to prevent pupae formation. 20 adult *C. sycophanta* were placed in clear plastic boxes (30 × 45 × 30 cm) with moist soil at the bottom. Adults were fed with *T. pityocampa* larvae regularly. After mating, *C. sycophanta* eggs were collected from the soil with a soft tip brush, and eggs were reserved in moist cotton until eggs cracked. Subsequently, *C. sycophanta* larvae were placed individually in clear, hard plastic cups (180 cm³) containing moist soil and covered with glass. One *T. pityocampa* larva per day was added to each cup for feeding. Predator insects were cultured in 65 (±5) % humidity, 25 (±5) °C, and 16:8 light: dark photoperiod.

*L. decemlineata* larvae were collected from a potato field in Kahramankazan district of Ankara province in 2022. The larvae were cultured in the laboratory under 60% (±10) humidity, 27(±2) °C temperature, and 16:8 light: dark photoperiod with potato plant leaves.

## Toxicity tests

Stock solutions of the molecules were dissolved in DMSO, AST-C peptide was dissolved in PBS supplemented with 0.1% BSA pH 7.4. Dilutions were prepared in PBS supplemented with 0.1% BSA pH 7.4 in 50, 100, 250, 500, 750, and 1000 ppm (mg/L) concentrations and later placed in an ultrasonic bath (Sonorex, Bandelin) for ~10 min at 37 °C until all precipitates had vanished. Toxicity tests were conducted 1 day after larvae collection. 5 µL of doses were applied on the dorsal surface of the head-prothorax junction topically. Dimilin 25 WP (%25 diflubenzuron), which the Ministry of Agriculture and Forestry of Türkiye suggests against *T. pityocampa*, was used as a positive control. Two negative control groups were prepared as PBS-applied and no application. Following the procedures, larvae were fed with 15 g *P. brutia* needles. Experiments were carried out on 10 larvae for each dosage, with three repeats.

## Side effect tests

Effective molecules of toxicity tests were applied on adults, 1st instar larvae of predator *C. sycophanta*, and 2nd instar larvae of potato beetle *L. decemlineata* to evaluate side effects.

*C. sycophanta* larvae and adults were placed individually in clear, hard plastic cups (180 cc) containing moist soil. 5 µL of 1000 ppm dose was applied topically on the head-prothorax junction's dorsal surface. 50 larvae and adults were used in the experiments. One *T. pityocampa* larva per day was added to each cup for feeding. Two negative controls were PBS-applied and no application groups.

5 µL of 1000 ppm dose were applied topically on the dorsal surface of the head-prothorax junction 2nd instar larvae of potato beetle. 30 larvae were used in assays, and larvae were placed on potato leaves inside the water-filled cups for feeding. Negative control groups were as previously described. Dimilin 25 WP (%25 diflubenzuron) was used as a positive control on *L. decemlineata*. Experiments were carried out with three repeats.

## Data collecting and analysis

Insect deaths were evaluated on the 3rd, 7th, and 14th days in toxicity and side effect experiments. The ones that were not feeding or not responding by a movement to stimulation with a soft-tipped brush were accepted as dead. Percentage mortality values generated on the 14th day were corrected using Abbott's formula[84]. Statistical analyses were executed with IBM SPSS Statistics (v26.0) and R Studio (2022.02.0), and two-way ANOVA and Tukey test were performed to detect the data's statistical significance. Lethal concentration values of $LC_{50-90}$ (dose that produces 50-90% mortality in the population) with 95% confidence intervals (CI) and slopes were obtained using PoloPlus software (LeOra Software, USA) for mortality values assessed on the 14th day for *T. pityocampa* and the 3rd day for *L. decemlineata*. The $LC_{50-90}$ values were considered to be similar if their 95% CIs overlapped ($P > 0.05$)[85]. Analysis between-subject effects on *T. pityocampa* larva length; one-way ANOVA tests assessing the impacts of molecule treatment on *T. pityocampa* larvae on larva length; and analysis of between-subjects effects on *T. pityocampa* head capsule width, alongside one-way ANOVA tests regarding the effects of molecule treatment on *T. pityocampa* larvae to head capsule width are detailed in the Source Data File (Source Data Supplementary Data Tables 1–4). Source data is available as Source Data File.

## Reporting summary

Further information on research design is available in the Nature Portfolio Reporting Summary linked to this article.

## Data availability

The PDB structures of the obtained docking poses of the ligands, .smi files of ligands, calculated physicochemical properties of compounds including PAINS, allometry data, 3D structures of used small molecule libraries, raw data of simulation interaction diagrams, and raw data of Figures and Tables are provided at https://zenodo.org/records/10866636. All remaining data supporting the findings of this study are available within the paper, its Supplementary Information, and Source Data File. Source data are provided with this paper.

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

## Acknowledgements

The numerical calculations reported in this paper were partially performed at The Scientific and Technological Research Council of Turkiye (TUBITAK) ULAKBIM, High Performance and Grid Computing Center (TRUBA resources). Computational part of this study was funded by Scientific Research Projects Commission of Bahçeşehir University. Project number: BAUBAP 2021.01.26. The project was also funded by COST Action Grant Number: CA18133 ERNEST and TUBITAK Grant Number: 119Z921. We would like to thank Dr. Alper Devrim Ozkan for his useful scientific discussions. The authors dedicated this research paper to the 100 anniversary of the Republic of Türkiye.

## Author contributions

S.D. and N.B.I. conceived and designed the study. K.K., M.B.D., A.E.A., A.Y., A.S., B.C. conducted experiments and analyzed data. K.K., S.D., and N.B.I. wrote the manuscript with editorial input from all authors.

## Competing interests

The authors declare no competing interests.
