## [Peer Review File · Nature Communications]

Discovering Allatostatin Type-C Receptor Targeted Specific Agonists through in silico, in vitro, and in vivo approachesREVIEWER COMMENTS

Reviewer #1 (Remarks to the Author):

This manuscript describes the identification of lead specific insecticidal compounds targeting an allatostatin receptor characterised from the pine processionary moth. This data provides significant advancement in the development of insecticidal compounds targeting an insect G-protein coupled receptor.

General

Overall the manuscript is well written with few grammatical errors. Some of the abbreviations require identifying (e.g. MM/GBSA) and abbreviations should not be used at the commencement of sentences.

Introduction.

First sentence (page 2): Pesticides have been a part of human life for decades and are conventionally used for controlling pest management to prevent diseases and protect crops.

Fifth paragraph (page 3): Reproduction and metamorphosis of insects, as well as behavioral attributes, diapause, and stress responses, are regulated by the JH, which is secreted from the corpora allata, and endocrine glands near the insect brain. Do you mean the corpora allata are endocrine glands near the insect brain?

IT would be useful to describe the different allatostatins in insects, there are members of all 3 types in Lepidoptera, although allatostatin-C is the only one that functions as an allatostatin in this group of insects.

Sixth paragraph (page 3)

Explain in more detail what "silk nests" are.

Results

Results are well presented and adequately described.

Discussion

Considering the comments below in the methods section, some comment on the uptake of these compounds the across the cuticle/cervix (membrane between head and thorax) is required.

Discussion around their bioavailability (stability/delivery) would be useful to assess their potential as insecticides.

Methods

Molecular, receptor screening and MD simulations are appropriate.

A number of questions arise over the methods used for toxicity studies.

Toxicity test (page 16).

Molecules were dissolved in PBS supplemented with 0.1% BSA and prepared in 50, 100, 250, 500, 750, and 1000 ppm (mg/L) concentrations. Toxicity tests were conducted one day after larvae collection. First, 5 μ L of doses were applied on the dorsal surface of the head-prothorax junction topically.

You state molecules were dissolved in PBS (no concentration or pH provide)

Topical applications for efficacy testing are normally applied in very small volumes (< 1 μ l) of a

solvent (e.g. pentan-3-one) or at the very least with a wetting agent. Buffer will not stay applied to the insect (due to hydrophobicity of cuticle and membranes). Aqueous solutions also take a long time to evaporate, adding further risk the applied droplet “falling off” the insect.

Furthermore, the amount you have applied (5 µl) is far too large, the insect would be “swamped” with this amount of liquid and most of the liquid would be absorbed onto the surface holding the insects.

Hence, the application method would appear to be greatly inaccurate.

The native allatostatin-C peptide identified from the pine processionary moth is identical to the one originally characterised from *Manduca sexta* and highly conserved in Lepidoptera. This peptide is very insoluble in water (and buffer) and usually is dissolved in a solvent or DMSO. You need to explain how you managed to dissolve this peptide in buffer alone.

Were the test solutions quantified after stock solutions were prepared? Synthesised peptides often contain salts and the weight does not always match the amount of peptide re-suspended in solution.

Reviewer #2 (Remarks to the Author):

Title: GPCR-Targeted Next-Generation Pesticides: A Combined *in silico*, *in vitro*, and *in vivo* Approach to Discover Novel Allatostatin Type-C Receptor Targeted Specific Agonists

By Kahveci et al.,

The study reports the allatostatin type-C receptor-targeted agonists to develop next-generation pesticides using the *in silico*, *in vitro* and *in vivo* methods. Authors selected the allatostatin type-C and its receptor, as the biological target, identified from the pine processionary moth (*Thaumetopoea pityocampa*), because the neuropeptide allatostatin functions to inhibit JH biosynthesis in insects.

Then *in silico* approach to virtual screening with MD simulation of the receptor using both GPCR and peptidomimetics libraries and screened four compounds. Followed the GPCR *in vitro* tests to measure EC50 values with HEK293 cells using Ca²⁺ mobilization in the TGF- α assay. Then tested the toxicities of each compound on the moth with topical application. And then, the side effect from the compounds was evaluated on the moth larvae. Overall, this is a good and novel approach for discovering new insecticides: targeting the neuropeptides and their receptors to screen agonists that are potential active ingredients. Particularly, three approaches, *in silico*, *in vitro*, and *in vivo*, are excellent methodology components. Although my expertise is not in the *in silico* methods, including virtual screening, MD simulation, and MM-GBSA analysis, I'm not opposing the manuscript to be published in the journal; however, some issues need to be clarified before acceptance.

Major questions: Authors showed EC50 values of the compounds that are agonistic, but all of these values are at very low concentrations (µM level). Can authors confirm or show any evidence whether these compounds inhibited JH biosynthesis after topical application on the moth (from their toxicity tests)? Because the biological function of the biological target is to inhibit the JH biosynthesis in this study.

Among EC50 values: AST-C followed D074-0013 (too low compared to the natural ligand); Mortality: AST-C followed V029-3547 (except for 3-day); LC50: AST-C followed V029-3547.

P12: Authors described, "Consequently, it was determined that D074-0034, J100-0311, and V029-3547 compounds ... can be used and developed as next-generation insecticides in AlstR-C-targeted

T.pityocampa control". From all the test results, AST-C is the most active and effective on the moth. Why can't AST-C be used for insecticide development? How can you explain or justify this? Please discuss more about this.

Minor issues:

Docking score: D074-0034 is the highest; how was the score of the AST-C? Also, the AST-C peptide sequence should be disclosed in the manuscript somewhere.

Page 11: "Since the effect of a molecule can be entirely different in a living organism, our investigations

continued with in vivo studies which are influential in uncovering unexpected or new mechanisms of pharmacological action" What is it meant? Describe be clear.

With the four compounds, their EC50 values, mortalities, and LC50 values are not consistent except for the natural ligand AST-C. Can you explain or discuss this?

Mortality against moth tested with the ligand and four agonists during three different periods, AST-C was the most effective (Fig 4a-c) – any statistically significant among them? Please indicate the statistical results.

P16: topical application – the AST-C and the compounds were dissolved in PBS, indicating that the peptide and the four compounds seem more hydrophilic rather than hydrophobic. How could the peptide and the agonists penetrate the moth larvae's cuticle (p15 toxicity test)?

P17: add the reference cited in the 1st paragraph.

Reviewer #3 (Remarks to the Author):

The manuscript "GPCR-Targeted Next-Generation Pesticides: A Combined in silico, in vitro, and in vivo Approach to Discover Novel Allatostatin Type-C Receptor Targeted Specific Agonists" describes the discovery of three potential novel pesticides against the pine processionary moth. The authors use in silico methods to determine candidate molecules to be tested in vitro and in vivo thereafter. Out of four tested molecules, all show agonistic behavior and all of them are also active in an in vivo assay against *T. pityocampa*, but not larvae of two other control insects.

While in principle the study seems to have been executed in a straightforward manner and the results -- if they can indeed be corroborated -- are exciting, my two main points of doubt are (a) the extraordinary hit rate and (b) the insufficient description of methods and results (basically all methods need more details). To provide more detail, I have listed points below in the order in which they appear in the manuscript. The results of these requests should be included in the manuscript.

* in the description of the docking screen, key details are missing in the text: How many molecules were docked in total? Were the two libraries docked separately or ranked together? Were really only the top-10 molecules chosen for further processing, no further selection criteria applied?

* as the docking calculations were based on the homology model of the receptor and the docked peptide, the complex needs to be provided together with the manuscript, in pdb or mol2 format.

* similarly, the docked poses of the hits in the receptor should be provided as pdb or mol2 files. 2D interaction depictions are not enough. Along the same lines, the SMILES codes of all molecules shown in the manuscript need to be listed in the SI. Also, all molecules should be run through PAINS filters (even though the vendor should already have done that) and the results reported.

* as any experiment, also MD simulations to assess complex stability need a negative control. Otherwise, the information provided by the analysis is limited. If the authors have access to an experimentally proven nonbinder of the AlstR-C, this would be ideal, otherwise a "random" molecule of similar biophysical properties but low 2D similarity would do. This is an important point, as the authors list this step as necessary in the workflow. In order for this to be true, the method needs to be able to discriminate between ligands and nonbinders in a statistically significant way.

* for the TGF-alpha shedding assay, it is unclear what the negative control was.

* as stated above, the hit rate of 100% strikes me as too good to be true. Such hit rates are extremely uncommon in primary virtual screens, even when docking is complemented by higher-level methods. That being said, the curves shown in Fig. 3 do rather fuel my doubts. The very steep EC50 curves of all test compounds are usually considered a hallmark of unspecific, aggregation-based inhibition. I acknowledge that the in vivo results are encouraging, but the authors are making the claim that the pesticidal activity comes from agonism at the AlstR-C -- and this is not sufficiently shown in my view. Therefore, the authors should do the following: (i) test their compounds on an unrelated receptor in identical assay conditions and compound concentrations. Unspecific effects can be buffer-specific, so the conditions need to match; (ii) test the compounds on the AlstR-C in an orthogonal in vitro assay to show that no other component of the TGF-alpha assay is inhibited; (iii) in case the "control" experiment is simply buffer without compound, the authors should include a physicochemically similar molecule that has low chances of binding to AlstR-C. Particularly the values for V029-3547 are so small that they could be rather unspecific. Ideally, that ligand would be the same as the one used as negative control in the MD simulations.

* the specificity assays in vivo (*C. sycophanta* and *L. decemlineata*), conversely, lack a positive control to demonstrate that these insects can be killed at all under the conditions used.

Points by points answers to the REVIEWER COMMENTS

Reviewer #1 (Remarks to the Author):

This manuscript describes the identification of lead-specific insecticidal compounds targeting an allatostatin receptor characterised by the pine processionary moth. This data provides a significant advancement in the development of insecticidal compounds targeting an insect G-protein coupled receptor.

General

Overall the manuscript is well written with few grammatical errors. Some of the abbreviations require identifying (e.g. MM/GBSA) and abbreviations should not be used at the commencement of sentences.

Answer: Fixed. Abbreviations have been revised to be used where they first appear in the text.

Introduction.

First sentence (page 2): Pesticides have been a part of human life for decades and are conventionally used for controlling pest management to prevent diseases and protect crops.

Answer: Fixed. It has been corrected in the revised manuscript.

Fifth paragraph (page 3): Reproduction and metamorphosis of insects, as well as behavioral attributes, diapause, and stress responses, are regulated by the JH, which is secreted from the corpora allata, and endocrine glands near the insect brain. Do you mean the corpora allata are endocrine glands near the insect brain?

Answer: Yes, the corpora allata is an endocrine gland that secretes Juvenile hormones. Detailed information has been added to the revised manuscript.

IT would be useful to describe the different allatostatins in insects, there are members of all 3 types in Lepidoptera, although allatostatin-C is the only one that functions as an allatostatin in this group of insects.

Answer: Further information on allatostatins is now provided in the revised manuscript.

Sixth paragraph (page 3)

Explain in more detail what “silk nests” are.

Answer: Silk nests are communal silk structures with defensive and thermoregulatory functions, sometimes described as bivouacs. Each nest may contain dozens to hundreds of caterpillars, and contributes to the damage they inflict on pine trees. The sentence in question is adjusted in the revised manuscript to clarify this.

Results

Results are well presented and adequately described.

Discussion

Considering the comments below in the methods section, some comment on the uptake of these compounds across the cuticle/cervix (membrane between head and thorax) is required.

Answer: References regarding pesticide administration on insect cuticles have been included in the revised manuscript. Briefly, the insect cuticle is composed of multiple layers, with exo- and endocuticular layers accounting for most of its thickness (and, correspondingly, attracting the greatest amount of scientific attention) (10.1016/j.asd.2004.05.006). While the prothorax is well-developed in insects such as cockroaches and beetles, where it forms a durable shield on the tergum, (10.1007/978-94-009-1189-5_2) the caterpillars used in our study are characterized by a soft and flexible cuticle that often lacks a significant exocuticular layer. (10.1007/0-306-48380-7_2198).

Later-instar caterpillars (particularly fifth- and sixth-instars) do possess considerably thicker cuticles, but the larvae used in our work belong to the second instar and are expected to readily absorb pesticides from a thin layer of cuticle. In addition, the administration of a liquid pesticide sample on the thorax of larval insects is common in the literature (references to support this view have been added to the discussion). Thus, we do not anticipate the cuticle to limit the delivery of our hit molecules in our pine processionary model. Discussion related to this part is detailed at the revised paper.

Discussion around their bioavailability (stability/delivery) would be useful to assess their potential as insecticides.

Answer: Thank you for your valuable feedback. We appreciate your suggestion to include a discussion on the bioavailability, stability, and delivery aspects of the identified small molecules in our study. We agree that such considerations are crucial in assessing the potential of these peptides as insecticides. In our revised manuscript, we incorporated a discussion on the bioavailability of the identified small molecules, emphasizing their stability within the insect system and the mechanisms employed for effective delivery. By addressing these aspects, we aim to provide a comprehensive understanding of the potential applications of these peptides as insecticides. We believe that this addition will enhance the overall impact and relevance of our study.

There are two focused libraries that we used in this study: ChemDiv's GPCR Targeted Library and ChemDiv's Peptidomimetic Library. Among selected top-10 hit compounds, only 1 compound (J100-0311) was from peptidomimetic library and the rest were from GPCR-Targeted library. A collection of small molecule compounds tailored to target

GPCRs stands as a potent asset in drug discovery, offering a pivotal resource for uncovering novel therapeutic agents. GPCRs, given their involvement in diverse physiological processes and diseases, constitute a substantial portion of current drug targets. Selected hits are drug-like small organic compounds. Regarding peptidomimetics, as the name implies, peptidomimetics are small organic molecules that mimic the action of peptides. While these molecules may exhibit structural similarities to peptides, they notably differ either in their side chains or molecular backbones. The primary components of the focused and targeted peptidomimetic small molecule library employed in this study include fragments of α -helices and mimetics of β - and γ -turns. These elements are derived from diverse combinatorial templates that have undergone modification with both flexible and rigid substituents. Derived and synthesized compounds are drug-like small organic compounds so they have high structural stabilities.

We conducted a further examination of potential metabolites for the identified four hit compounds. To achieve this, we utilized the MetaCore/MetaDrug program from Clarivate Analytics. (<https://portal.genego.com/>) This feature allows us to exclusively predict metabolites for the investigated hit compound by applying specific metabolic cleavage and prioritization rules. All metabolic reaction types are selected, including prioritization and second-pass options. The three potential metabolites of the hit compounds were then subjected to docking at the binding pocket of AlstR-C using same parameters with the original molecules, and their docking scores were compared with each hit. (Table R1) The results show mainly similar docking scores for the metabolites except one of the metabolites of the V029-3547, suggesting that in the case of metabolite formation after oral administration due to enzymatic action, the biological effect of these hit compounds may not decrease significantly.

	2D Structure	Docking Score (kcal/mol)
D074-0013	 The structure of D074-0013 features a central imidazole ring. One nitrogen of the imidazole is bonded to a carbonyl group, which is further connected to a cyclopentane ring containing an oxygen atom. The other nitrogen of the imidazole is bonded to a phenyl ring with a hydroxyl group at the para position. The 2-position of the imidazole ring is substituted with a phenyl ring that has a phenoxy group at the para position.	-8.981
Metabolite-1 of D074-0013	 The structure of Metabolite-1 of D074-0013 is a modified version of the parent compound. It features a central imidazole ring with a hydroxyl group at the 2-position. One nitrogen of the imidazole is bonded to a carbonyl group, which is further connected to a cyclopentane ring containing an oxygen atom. The other nitrogen of the imidazole is bonded to a phenyl ring with a hydroxyl group at the para position. The 5-position of the imidazole ring is substituted with a phenyl ring that has a phenoxy group at the para position.	-8.752

Metabolite-2 of D074-0013		-8.431
Metabolite-3 of D074-0013		-8.854
D074-0034		-9.090
Metabolite-1 of D074-0034		-8.396
Metabolite-2 of D074-0034		-7.546
Metabolite-3 of D074-0034		-8.602

V029-3547		-8.418
Metabolite-1 of V029-3547		-7.951
Metabolite-2 of V029-3547		-7.524
Metabolite-3 of V029-3547		-5.639
J100-0031		-8.497
Metabolite-1 of J100-0031		-8.403
Metabolite-2 of J100-0031		-7.762

Metabolite-3 of J100-0031		-8.252
---	--------

Table R1: 2D structures and docking scores of one of the identified hit compounds and their potential metabolites.

Peptides or small organic molecules may undergo degradation due to environmental factors like sunlight, heat, and enzymes, potentially diminishing the active peptide reaching the intended pests and impacting bioavailability. It is crucial to formulate these active compounds into a stable and efficient product to enhance their bioavailability.

Adequate formulation serves to safeguard from degradation and enhances their delivery to the intended organisms. The choice of application method for small organic compounds or peptides in the target area can also impact their bioavailability. Factors such as spraying, injection, or soil application can influence the efficacy with which these hits reach and interact with pests.

Peptides need to maintain stability within the insect's body before showcasing their bio-functionalities. For instance, C-type allatostatin, when orally administered to lepidopteran larvae, does not exhibit activity as it rapidly breaks down in the digestive system due to enzymatic action (10.1002/arch.20265). However, it was shown that substitution of Arginine residues by the D-isomer prevented hydrolysis of the peptide by the cathepsin L-like cysteine and trypsin-like proteases present in the aphid gut extract (10.1016/j.peptides.2009.06.017). This modification also enhances the half-life, showing a 2.6-fold increase in vitro compared to the native peptides. This improved stability renders the modified peptide more robust and potent than its native counterpart. Besides chemical modification, the encapsulation of bioactive peptides in microcapsules and liposomes is recognized as a promising technology to enhance metabolic stability (10.1016/j.crf.2022.10.031, 10.1039/d1ra08590e). Studies have demonstrated that nanodiamonds coupled with Neb-colloostatin can permeate the cuticle and disrupt the cellular and humoral immune responses across all developmental stages of *Tenebrio molitor* (10.1038/s41598-021-97924-x). Therefore, for further studies, in case we need to improve metabolic stability and bioavailability, different pharmaceutical technological approaches will be considered.

Methods

Molecular, receptor screening, and MD simulations are appropriate. A number of questions arise over the methods used for toxicity studies.

Toxicity test (page 16).

Molecules were dissolved in PBS supplemented with 0.1% BSA and prepared in 50, 100,

250, 500, 750, and 1000 ppm (mg/L) concentrations. Toxicity tests were conducted one day after larvae collection. First, 5 μ L of doses were applied on the dorsal surface of the head-prothorax junction topically.

You state molecules were dissolved in PBS (no concentration or pH provided)

Answer: We apologize for the lack of clarity for the solution preparation. Stock solutions of the hit compounds were dissolved in DMSO, while the AST-C peptide was dissolved in PBS supplemented with 0.1% BSA pH 7.4 (allatostatins are water-soluble). In the TGF α shedding assay, dilutions are made with HBSS. In the Glo-sensor assay, dilutions are made with beetle luciferin 2mM/Leibovitz's L-15 Medium, no phenol red 1% FBS solutions. For in vivo assays, dilutions are made with PBS supplemented with 0.1% BSA pH 7.4 and later placed in an ultrasonic bath (Sonorex, Bandelin) for ~10 min at 37°C until all precipitates had vanished. This information is now included in the revised manuscript.

Topical applications for efficacy testing are normally applied in very small volumes (< 1 μ l) of a solvent (e.g. pentan-3-one) or at the very least with a wetting agent. Buffer will not stay applied to the insect (due to hydrophobicity of cuticle and membranes). Aqueous solutions also take a long time to evaporate, adding further risk of the applied droplet “falling off” the insect. Furthermore, the amount you have applied (5 μ l) is far too large, the insect would be “swamped” with this amount of liquid, and most of the liquid would be absorbed onto the surface holding the insects.

Hence, the application method would appear to be greatly inaccurate.

Answer: When larval cuticles are densely covered with hairs, there is a potential limitation where xenobiotics applied may not be absorbed entirely. (10.1038/s41598-021-95284-0) To address this, the peptide application volume was increased to 5 μ l to ensure penetration of the larval integument. During application, the pipette tip was gently touched to the larval surface, and post-application, careful examination was conducted to verify droplet contact with the insect surface. The presence of hairs also aided in preventing droplet detachment.

In addition, 5 μ l volume did not cause larvae to drown as can be seen in the mortality of the control group (<%10). Also, the majority of the mortality in the treatment group would be in the very first days. But it can be seen that %mean mortality from the 7th and 14th day was much higher than the 3rd day after application.

It must be noted that the large volume of application was necessitated by the correspondingly large size (even at second instar) and dense cuticular setae of the pine processionary caterpillar, which may limit pesticide absorption. We now provide photographs of the solution application process (Figure S6, supplementary materials) to support our claim that excess solution is not repelled or wiped onto the substrate, and provide references using similarly large (2-5 μ L) volumes. Nevertheless, we agree with the reviewer that the volume is uncharacteristically large and mention loss of solution as a possible disadvantage of our method.

The native allatostatin-C peptide identified from the pine processionary moth is identical to the one characterised initially from *Manduca sexta* and highly conserved in Lepidoptera. This peptide is very insoluble in water (and buffer) and usually is dissolved in a solvent or DMSO. You need to explain how you managed to dissolve this peptide in the buffer alone.

Were the test solutions quantified after stock solutions were prepared? Synthesised peptides often contain salts and the weight does not always match the amount of peptide re-suspended in solution.

Answer: To fully dissolve the AST-C in PBS, solvent supplemented with 0.1% BSA (pH 7.4) prepared in 50, 100, 250, 500, 750, and 1000 ppm (mg/L) concentrations and later placed in an ultrasonic bath (Sonorex, Bandelin) for ~10 min at 37°C until all precipitates had vanished. AST-C solutions were quantified in Thermo Scientific™ NanoDrop™ 2000/2000c Spectrophotometers.

This part is clarified in the revised paper.

Reviewer #2 (Remarks to the Author):

Title: GPCR-Targeted Next-Generation Pesticides: A Combined in silico, in vitro, and in vivo Approach to Discover Novel Allatostatin Type-C Receptor Targeted Specific Agonists
By Kahveci et al.,

The study reports the allatostatin type-C receptor-targeted agonists to develop next-generation pesticides using the in silico, in vitro and in vivo methods. Authors selected the allatostatin type-C and its receptor, as the biological target, identified from the pine processionary moth (*Thaumetopoea pityocampa*), because the neuropeptide allatostatin functions to inhibit JH biosynthesis in insects. Then in silico approach to virtual screening with MD simulation of the receptor using both GPCR and peptidomimetics libraries and screened four compounds. Followed the GPCR in vitro tests to measure EC50 values with HEK293 cells using Ca²⁺ mobilization in the TGF- α assay. Then tested the toxicities of each compound on the moth with topical application. And then, the side effect from the compounds was evaluated on the moth larvae. Overall, this is a good and novel approach for discovering new insecticides: targeting the neuropeptides and their receptors to screen agonists that are potential active ingredients. Particularly, three approaches, in silico, in vitro, and in vivo, are excellent methodology components. Although my expertise is not in the in silico methods, including virtual screening, MD simulation, and MM-GBSA analysis, I'm not opposing the manuscript to be published in the journal; however, some issues need to be clarified before acceptance.

Major questions: Authors showed EC50 values of the compounds that are agonistic, but all of these values are at very low concentrations (μ M level). Can authors confirm or show any evidence whether these compounds inhibited JH biosynthesis after topical application on the moth (from their toxicity tests)? Because the biological function of the biological target is to inhibit the JH biosynthesis in this study.

Answer: Juvenile hormone (JH) is a key regulator in the development and metamorphosis of insects. It plays a crucial role in controlling the transition between different life stages, such as from larval to pupal stages and from pupal to adult stages in insects that undergo complete metamorphosis. In the larval stage, the JH promotes the maintenance of the growth and differentiation of larval tissues. It ensures that the insect grows and molts into larger instars before transitioning to the pupal stage. It also plays a role in developing and maintaining adult reproductive organs and behaviors. It influences the maturation of reproductive structures and the onset of reproductive activities. JH allows insects to adapt their development to environmental conditions, ensuring that the timing of metamorphosis is well-suited to the prevailing ecological circumstances. The treatment with the compounds leads to decreased larva size (from head to cauda) and head capsule width. Failures in growth, molting and pupation are primary consequences of JH inhibition. However, such evidence is indirect. Related results are added now as a new figure (Figure 6) and Tables S3, S4, S5, S6 at the revised paper.

Inhibition of JH via the allatostatin pathway can lead to feeding suppression (10.1016/j.ygcen.2008.08.003, 10.1046/j.0307-6962.2001.00233.x). Discussion about how JH inhibition can affect larvae has been highlighted in the revised paper. Observations about the behavioral changes like a decrease in feeding and defecation, deterioration in netting behavior, developmental retardation, and the absence of the characteristic "pine processionary larvae marching" behavior in the T. pityocampa larvae are highlighted in the manuscript.

Among EC50 values: AST-C followed D074-0013 (too low compared to the natural ligand); Mortality: AST-C followed V029-3547 (except for 3-day); LC50: AST-C followed V029-3547.

P12: Authors described, "Consequently, it was determined that D074-0034, J100-0311, and V029-3547 compounds ... can be used and developed as next-generation insecticides in AlstR-C-targeted T.pityocampa control". From all the test results, AST-C is the most active and effective on the moth. Why can't AST-C be used for insecticide development? How can you explain or justify this? Please discuss more about this.

Answer: As a peptide, AST-C has a relatively short shelf life and is quickly degraded in the environment. In addition, they can not be administered orally as they would be digested. While there are efforts to stabilize allatostatins (e.g. via D-isomer substitution), small organic compound ligands such as peptidomimetics offer a more stable alternative with the potential for cheaper mass production efforts. Our manuscript takes a similar path by using GPCR-targeted and peptidomimetic compounds instead of peptides, which allows the design of pesticides with mass production capacity and longer shelf-lives (as well as half-lives in vivo), while retaining selectivity against their targets. This information is now included in the discussion section.

Minor issues:

Docking score: D074-0034 is the highest; how was the score of the AST-C? Also, the AST-C peptide sequence should be disclosed in the manuscript somewhere.

Answer: We employed Glide Peptide docking to assess and compare the docking scores of AST-C with identified hit compounds. The top-docking score for AST-C was determined as -8.816 kcal/mol. The docking score for D074-0034 was calculated to be -9.090 kcal/mol. When considering the molecular sizes of D074-0034 and AST-C peptide, the ligand efficiency score (calculated as docking score per the number of heteroatoms) for D074-0034 was determined as -0.252 kcal/mol, while the corresponding ligand efficiency score for AST-C peptide was calculated as -0.066 kcal/mol. The following figure shows the top-docking pose of AST-C (green colored) at the AlstR-C. (Figure R1)

Figure R1: Top-docking pose of native AST-C peptide (represented with green colored structure) at the binding pocket of the AlstR-C. Docking score of AST-C peptide: -8.816 kcal/mol.

Page 11: "Since the effect of a molecule can be entirely different in a living organism, our investigations continued with in vivo studies which are influential in uncovering unexpected or new mechanisms of pharmacological action" What is it meant? Describe be clear.

Answer: *We had meant that in silico and in vitro studies are by themselves insufficient for determining the effects of a pesticide, and in vivo tests must be conducted to determine whether e.g. the insect might survive an otherwise promising pesticide through an unexpected rescue pathway. The manuscript has been amended accordingly.*

With the four compounds, their EC₅₀ values, mortalities, and LC₅₀ values are not consistent except for the natural ligand AST-C. Can you explain or discuss this? Mortality against moth tested with the ligand and four agonists during three different periods, AST-C was the most effective (Fig 4a-c) – any statistically significant among them? Please indicate the statistical results.

Answer: In terms of mortality against moths during three different periods, Figure 4a-c illustrates that AST-C was the most effective among the applied ligands. To ascertain the statistical significance among them, additional statistical tests are conducted. We included detailed statistical results in our revised manuscript to provide a comprehensive understanding of the significance of the observed differences in mortality. It's essential to note that while the native ligand may exhibit superior binding affinity, it could also have limitations such as a short-lived duration of action and lack of specificity. For these reasons, the development of small-molecule synthetic organic compounds becomes imperative, as they offer the potential for increased stability, prolonged effectiveness, and improved specificity in modulating the targeted biological processes. This perspective underscores the significance of exploring synthetic compounds as promising alternatives to address the shortcomings associated with natural ligands. This additional analysis will contribute to a more thorough and accurate interpretation of the experimental outcomes.

P16: topical application – the AST-C and the compounds were dissolved in PBS, indicating that the peptide and the four compounds seem more hydrophilic rather than hydrophobic. How could the peptide and the agonists penetrate the moth larvae's cuticle (p15 toxicity test)?

*Answer: We would like to thank the reviewer for this insightful comment. The insect cuticle comprises multiple layers, and advancements in comprehending its mechanical properties have predominantly focused on the exo- and endocuticular layers, constituting the majority of its thickness (10.1016/j.asd.2004.05.006). Notably, the prothorax is highly developed in insects like cockroaches and beetles, forming a substantial shield (10.1007/978-94-009-1189-5_2). However, many lepidopteran larvae exhibit a soft, flexible cuticle with minimal or no exocuticle (10.1007/0-306-48380-7_2198). Furthermore, the cuticles of later instars in lepidopteran larvae (fifth- and sixth-instar) are notably thicker than those of third- and fourth-instar larvae, with older larvae demonstrating increased tolerance to insecticides. In our study, we focused on second-instar larvae, which are anticipated to have thinner cuticles. Therefore, we did not anticipate the cuticle in these second-instar larvae of *T. pityocampa* to present challenges in peptide delivery. We have now included this information in the manuscript.*

P17: add the reference cited in the 1st paragraph.

Answer: Fixed.

Reviewer #3 (Remarks to the Author):

The manuscript "GPCR-Targeted Next-Generation Pesticides: A Combined in silico, in vitro, and in vivo Approach to Discover Novel Allatostatin Type-C Receptor Targeted Specific Agonists" describes the discovery of three potential novel pesticides against the pine processionary moth. The authors use in silico methods to determine candidate molecules to be tested in vitro and in vivo thereafter. Out of four tested molecules, all show agonistic behavior and all of them are also active in an in vivo assay against *T. pityocampa*, but not larvae of two other control insects.

While in principle the study seems to have been executed in a straightforward manner and the results -- if they can indeed be corroborated -- are exciting, my two main points of doubt are (a) the extraordinary hit rate and (b) the insufficient description of methods and results (basically all methods need more details). To provide more detail, I have listed points below in the order in which they appear in the manuscript. The results of these requests should be included in the manuscript.

Answer a) This research builds upon a prior doctoral thesis conducted within our group (PhD thesis of Aida Shahraki, "Functional And Structural Insights Into A Novel Insect G Protein-Coupled Receptor, Allatostatin Receptor Type C Of Pine Processionary Moth". In this earlier study, only a single molecule from the ligand library (Otava beta-turn peptidomimetics library) demonstrated agonist potential. Consequently, we pursued this investigation using diverse libraries. The in vitro results of the tested molecules are illustrated in the figure down below. (Figure is from PhD thesis of Aida Shahraki) Additional information and references related to the thesis has been incorporated and cited in the revised manuscript.

Shahraki, A. (2020). *Functional And Structural Insights Into A Novel Insect G Protein-Coupled Receptor, Allatostatin Receptor Type C Of Pine Processionary Moth* (Publication No. 645040) [Doctoral dissertation, Bogazici University]. Ulusal Tez Merkezi. <https://tez.yok.gov.tr/> and <http://digitalarchive.boun.edu.tr/handle/123456789/15521?locale-attribute=en>

Figure 5.32: G protein activation assay performed for the finally selected seven compounds.

Answer b: We'd like to thank the reviewer for this insightful comment, in the revised paper more details about the used methods have been provided.

* in the description of the docking screen, key details are missing in the text: How many molecules were docked in total? Were the two libraries docked separately or ranked together? Were really only the top-10 molecules chosen for further processing, no further selection criteria applied?

Answer: The number of molecules used in the docking is provided now in the Material and Methods section of the revised paper. The requested details were updated in the text. Two libraries were ranked together and besides the docking scores, functional groups of the molecules were compared. This part is clarified in the revised paper.

* as the docking calculations were based on the homology model of the receptor and the docked peptide, the complex needs to be provided together with the manuscript, in pdb or mol2 format.

Answer: All the complex structures are provided as datasets in the revised paper. Furthermore, we upload all these complex structures and make them available at zenodo now: <https://zenodo.org/records/10570730>

* similarly, the docked poses of the hits in the receptor should be provided as pdb or mol2 files. 2D interaction depictions are not enough. Along the same lines, the SMILES codes

of all molecules shown in the manuscript need to be listed in the SI. Also, all molecules should be run through PAINS filters (even though the vendor should already have done that) and the results reported.

Answer: The docking poses are provided as pdb (<https://zenodo.org/records/10570730>). The table involving the SMILES codes is added to the Supplementary Data (Table S2) and the physicochemical property calculations including PAINS filter results are added to SI. The findings indicate that, with the exception of compound C794-1617 (not considered in our in vitro and in vivo experiments due to its comparatively poorer MM/GBSA scores), the remaining nine compounds do not exhibit any PAINS-related issues.

* as any experiment, also MD simulations to assess complex stability need a negative control. Otherwise, the information provided by the analysis is limited. If the authors have access to an experimentally proven nonbinder of the AlstR-C, this would be ideal, otherwise a "random" molecule of similar biophysical properties but low 2D similarity would do. This is an important point, as the authors list this step as necessary in the workflow. In order for this to be true, the method needs to be able to discriminate between ligands and nonbinders in a statistically significant way.

Answer: Since there is no experimentally proven non-binder of the AlstR-C, decoy molecules were generated, and MD simulations and MM/GBSA analysis were applied. (Figure S1, Table S1) Our results show that none of the decoy compounds exhibit superior docking or average MM/GBSA scores in comparison to the compounds we tested.

* for the TGF-alpha shedding assay, it is unclear what the negative control was.

* as stated above, the hit rate of 100% strikes me as too good to be true. Such hit rates are extremely uncommon in primary virtual screens, even when docking is complemented by higher-level methods. That being said, the curves shown in Fig. 3 do rather fuel my doubts. The very steep EC50 curves of all test compounds are usually considered a hallmark of unspecific, aggregation-based inhibition. I acknowledge that the in vivo results are encouraging, but the authors are making the claim that the pesticidal activity comes from agonism at the AlstR-C -- and this is not sufficiently shown in my view. Therefore, the authors should do the following: (i) test their compounds on an unrelated receptor in identical assay conditions and compound concentrations. Unspecific effects can be buffer-specific, so the conditions need to match; (ii) test the compounds on the AlstR-C in an orthogonal in vitro assay to show that no other component of the TGF-alpha assay is inhibited; (iii) in case the "control" experiment is simply buffer without compound, the authors should include a physicochemically similar molecule that has low chances of binding to AlstR-C. Particularly the values for V029-3547 are so small that they could be rather unspecific. Ideally, that ligand would be the same as the one used as negative control in the MD simulations.

Answer: The relevant experiments have been performed and included in the revised manuscript. Briefly, we have repeated the TGF α shedding assay with the serotonin receptor (5-HTR4) and observed no difference between AST-C and the hit compounds.

5-HTR4 was used in the TGF α shedding assay as a negative control since it was already available in our lab with its natural ligand (Serotonin). We wanted to test whether the identified hit compounds will affect the 5-HTR4 receptor as the AlstR-C. A Glo-sensor cAMP inhibition assay was used to orthogonally confirm that both AST-C and the tested hit compounds target AlstR-C (we use an initial Forskolin upregulation step as AlstR-C is G α_i -coupled). Since AlstR-C is G α_i coupled, a decrease in cAMP levels was expected. The real-time measurement results and related methods information are added and highlighted in the manuscript.

* the specificity assays in vivo (*C. sycophanta* and *L. decemlineata*), conversely, lack a positive control to demonstrate that these insects can be killed at all under the conditions used.

*Answer: The missing experiment was performed with *L. decemlineata* only, as *C. sycophanta* could not be collected due to its seasonal reproduction cycle. Diflubenzuron (Dimilin) was used as a positive control; its LC₅₀ graph is added to Figure 5.*

REVIEWERS' COMMENTS

Reviewer #1 (Remarks to the Author):

The manuscript has been revised and my comments addressed as requested.

One minor change is still required to additional text added:

Page 13, last paragraph, statements require supporting evidence (citations).

As a peptide, AST-C exhibits a comparatively short shelf life and undergoes rapid degradation in the environment. - Reference required to support this.

Furthermore, administering it orally is impractical due to digestion concerns. Reference required Although attempts have been made to enhance the stability of allatostatins through methods like D-isomer substitution, small organic compound ligands such as peptidomimetics present a more stable alternative with the potential for cost-effective mass production efforts. - reference(s) required.

Reviewer #2 (Remarks to the Author):

Most of the questions have been addressed in the revised manuscript.

Reviewer #3 (Remarks to the Author):

I have re-reviewed the manuscript "GPCR-Targeted Next-Generation Pesticides: A Combined in silico, in vitro, and in vivo Approach to Discover Novel Allatostatin Type-C Receptor Targeted Specific Agonists" and found that the authors have adequately addressed all my previous concerns.

In particular, the appropriate control experiments, both in silico and in vivo, have been carried out and strengthen the claims made in the first version of the paper.

I also had a look at the pdb files that were provided and the docking poses of the 4 molecules that turned out to be ligands are adequate (which can't be said for all poses, but then these molecules did not bind, so this is consistent).

Reviewer #3 (Remarks on code availability):

Not applicable.

Points by points answers to the REVIEWER COMMENTS

REVIEWERS' COMMENTS

Reviewer #1 (Remarks to the Author):

The manuscript has been revised and my comments addressed as requested.

Response: We thank the Reviewer for her/his positive feedback and insightful comments.

One minor change is still required to additional text added:
Page 13, last paragraph, statements require supporting evidence (citations).

Response: The citations have been added.

As a peptide, AST-C exhibits a comparatively short shelf life and undergoes rapid degradation in the environment. - Reference required to support this.

Response: The citation has been added.

Furthermore, administering it orally is impractical due to digestion concerns.

Response: The citation has been added.

Although attempts have been made to enhance the stability of allatostatins through methods like D-isomer substitution, small organic compound ligands such as peptidomimetics present a more stable alternative with the potential for cost-effective mass production efforts. - reference(s) required.

Answer: The citations have been added.

Reviewer #2 (Remarks to the Author):

Most of the questions have been addressed in the revised manuscript.

Response: We thank the Reviewer for her/his positive feedback and insightful comments.

Reviewer #3 (Remarks to the Author):

I have re-reviewed the manuscript "GPCR-Targeted Next-Generation Pesticides: A Combined in silico, in vitro, and in vivo Approach to Discover Novel Allatostatin Type-C Receptor Targeted Specific Agonists" and found that the authors have adequately addressed all my previous concerns.

In particular, the appropriate control experiments, both in silico and in vivo, have been carried out and strengthen the claims made in the first version of the paper.

I also had a look at the pdb files that were provided and the docking poses of the 4 molecules that turned out to be ligands are adequate (which can't be said for all poses, but then these molecules did not bind, so this is consistent).

Response: We thank the Reviewer for her/his positive feedback and insightful comments.

Reviewer #3 (Remarks on code availability):

Not applicable.